# Label-free imaging and classification of live *P. falciparum* enables high performance parasitemia quantification without fixation or staining

**Paul Lebel**[1]*, **Rebekah Dial**[2¤], **Venkata N. P. Vemuri**[1], **Valentina Garcia**[2], **Joseph DeRisi**[1,2], **Rafael Gómez-Sjöberg**[1]

**1** Chan Zuckerberg Biohub, San Francisco, California, United States of America, **2** Department of Biochemistry and Biophysics, University of California San Francisco, San Francisco, California, United States of America

¤ Current address: Denali Therapeutics, South San Francisco, California, United States of America
* paul.lebel@czbiohub.org

**Data Availability Statement:** The data is accessible on the BioStudies database under the following accession numbers: UV Raw dataset:

## Abstract

Manual microscopic inspection of fixed and stained blood smears has remained the gold standard for *Plasmodium* parasitemia analysis for over a century. Unfortunately, smear preparation consumes time and reagents, while manual microscopy is skill-dependent and labor-intensive. Here, we demonstrate that deep learning enables both life stage classification and accurate parasitemia quantification of ordinary brightfield microscopy images of live, unstained red blood cells. We tested our method using both a standard light microscope equipped with visible and near-ultraviolet (UV) illumination, and a custom-built microscope employing deep-UV illumination. While using deep-UV light achieved an overall four-category classification of *Plasmodium falciparum* blood stages of greater than 99% and a recall of 89.8% for ring-stage parasites, imaging with near-UV light on a standard microscope resulted in 96.8% overall accuracy and over 90% recall for ring-stage parasites. Both imaging systems were tested extrinsically by parasitemia titration, revealing superior performance over manually-scored Giemsa-stained smears, and a limit of detection below 0.1%. Our results establish that label-free parasitemia analysis of live cells is possible in a biomedical laboratory setting without the need for complex optical instrumentation. We anticipate future extensions of this work could enable label-free clinical diagnostic measurements, one day eliminating the need for conventional blood smear analysis.

## Author summary

Although qualitative rapid diagnostic tests (RDTs) and PCR-based assays for malaria detection have existed for many years, in most malaria-endemic countries manual counting by microscopy remains the dominant modality for assessment of infection. This method involves smearing, fixation, staining, and manual inspection of blood smears—a practice that is time-consuming, labor-intensive, error-prone, and has varied little in over

https://www.ebi.ac.uk/biostudies/studies/S-BSST559 Leica Raw dataset: https://www.ebi.ac.uk/biostudies/studies/S-BSST567 Github repos for the software: https://github.com/czbiohub/UVScope-control https://github.com/czbiohub/Label-Free-Malaria https://github.com/czbiohub/luminoth-uv-imaging/.

**Funding:** This work was funded by the Chan Zuckerberg Biohub. The funders had no role in study design, data collection and analysis, decision to publish, or preparation of the manuscript. JD, RGS, PL, and VNPV earn salaries from Chan Zuckerberg Biohub.

**Competing interests:** I have read the journal's policy and the authors of this manuscript have the following competing interests: The authors declare provisional patent application 63/072,037 filed on 08/28/2020.

a century. Similarly, for laboratories around the world that grow *P. falciparum* for research, the process of assessing different stages of parasite growth is a daily ritual.

Here, we provide rigorous evidence that live, unstained parasites can be automatically distinguished and sub-categorized from a healthy background in the context of laboratory cell culture, by applying deep learning to ordinary microscopy images, with high-sensitivity and low false-positive rates. This is especially true for ring-stage parasites, which represent the dominant circulating form of parasite in humans and are notoriously difficult to detect and classify by traditional methods. We define the performance envelopes of these classification methods utilizing both a custom-built deep-UV microscope and a standard commercial brightfield microscope across a broad set of parameters. We also show that our machine classifiers are more accurate over a larger range of parasitemia than human technicians scoring Giemsa-stained smears prepared from the same samples.

This is a *PLOS Computational Biology* Methods paper.

## Introduction

Historically, malaria is among the deadliest infectious diseases in human history. Of the five *Plasmodium* species that cause illness in humans, *P. falciparum* accounts for 99% of cases in Africa, and 94% of all malaria deaths [1]. Not only is it distributed across many regions of the world, but its eradication is intimately linked to the complexity of its mosquito vector and the human population it infects. If left untreated, malaria causes severe febrile illness or death, especially in children and infants. The World Health Organization reported an estimated 228 million cases in 2018, and an estimated 405,000 deaths globally [1], the large majority of which occur in low-resource areas with little access to healthcare, representing some of the most vulnerable communities in the world.

Although anti-malarial drugs have been employed with success for centuries, chemoresistance remains an ongoing threat as strains evolve in response to widespread treatment regiments [2]. Despite the effectiveness of chloroquine (CQ), the mass administration of sub-curative doses of the drug in Cambodia in 1955 has been credited with the initial evolution of a CQ-resistant strain [2], with resistance also appearing independently in South America in the 1960s [3]. Artemisinin-based Combination Therapies (ACT) are highly effective, and thought to be less susceptible to resistance due to the combination of drugs with distinct mechanisms of action [4]. Unfortunately, front line artemisinin resistance has now been documented in both Southeast Asia and Latin America [4, 5]. Further, *P. falciparum* also possesses a well-evolved ability to evade host immunity in both the Anopheles vector and the human host through various allelic adaptations and antigenic variation [6]. These factors put considerable pressure on the therapeutic development process to continuously identify and target orthogonal mechanisms of action.

Ongoing investigations into topics such as drug-resistance and host immune evasion hinges on the ability to cultivate strains of *P. falciparum* in a laboratory setting. An essential part of *ex vivo* culturing includes daily assessments of the parasitemia and quantification of the life stages. While methods exist for flow-cytometry assisted quantification of parasitemia, manual counting remains the gold standard for evaluating these factors [7]. For each culture, a smear must be prepared which typically involves: blood smearing, drying, fixation, drying again, staining, and rinsing. On average, the staining process alone consumes 45 minutes [7]. Subsequently, several hundred or even thousands of cells must be inspected manually under the

microscope in order to quantify parasite stages with sufficient sampling power, consuming 15–30 minutes for a trained microscopist [7, 8]. Despite publication of standardized training methods, testing, and cross-validation of microscopist competence [9], errors in manual microscopy are likely [8], since counting depends on stain quality and technician skill level, and can be hampered by fatigue and reagent quality. The entire end-to-end manual procedure is estimated to be carried out hundreds of thousands of times annually [7].

The fields of label-free imaging and deep learning are both rapidly advancing [10–12], with a promising convergence in applications related to automated detection of blood pathogens. Label-free methods such as Differential Interference Contrast [13] and Optical Diffractive Tomography [14] have succeeded in producing detailed maps of RBCs with sufficient resolution for parasite detection and feature extraction. While they represent important advances, neither of these methods have yielded classification results sufficient for routine laboratory practice or diagnostic settings. On the other hand, quantitative phase microscopy (QPM) has been used to achieve high accuracy differentiation between late parasite stages and healthy cells, but lacks the ability to distinguish parasite stages from each other, and further, the method was not tested on ring stages—the most prevalent stage found in peripheral blood [15]. Digital Holographic microscopy has succeeded in achieving high-sensitivity parasite detection by combining complex microscopy instrumentation with specialized microfluidic devices [16]. Meanwhile, there are a growing number of efforts to process conventional Giemsa-stained blood smear images using deep learning [7, 17–24], including the use of Fourier Ptychographic Microscopy (FPM) with extended depth of focus and numerical aperture [25–27]. Notably, low-cost mobile devices and computation hardware have been leveraged to efficiently process thick blood smears [24]. These recent efforts have achieved high levels of performance by training deep networks to classify stained images. These also represent important contributions to the field, because although fixation, staining, and manual microscopy must still be performed, technician time spent inspecting images could potentially be eliminated. Automated, low-cost slide scanning has been recently achieved and combined with embedded computing hardware for image analysis [28]. This advance automates the labor-intensive microscopy portion, but still requires time-consuming fixation and staining prior to imaging, and sophisticated scanning stages. Other non-imaging label-free methods have exploited hemozoin crystallization by the parasite [29], or antibody-mediated electrochemical detection [30]. Others have exploited the mechanical hardening of infected RBCs using micro-fluidic strain sensors [31, 32], yielding up to 92% overall accuracy on binary infection calling, although it lacks clear lifecycle stage breakdown and comes at the cost of complex microfabrication techniques.

The absence of label-free parasite classification of ordinary brightfield microscopy images is likely the result of a century-long preconception that the interaction between visible light and biological matter is too weak (insufficient contrast), especially at sub-micron morphological features (insufficient resolution). Additionally, because the prevalence of parasites (parasitemia) is low compared to healthy cells, the performance requirement for such a technique to be useful is high, imposing stringent requirements on the maximum FPR, precision, and recall. To explore this topic we surveyed the wavelength dependence, from deep-ultraviolet (deep UV) to visible light, of image classification performance scoring parasite infection stages (healthy, ring, trophozoite, or schizont) in live, cultured red blood cells. In doing so, we demonstrated that the application of automated classification is possible over a broad spectrum of wavelengths, and that the combination of higher resolution and contrast at shorter wavelengths yields more clearly-resolved parasite physiology as compared with visible light. Additionally, we employed post-processing techniques that further improve upon raw classifier results, and by extensive parasitemia titration show that our calibrated machine classifiers

exceed the performance of manually-counted Giemsa-stained thin blood smears, in a non-clinical laboratory setting.

Deep UV imaging methods have previously been developed for use in biological imaging. It has been shown, for example, that cellular dry mass can be recorded from fixed cells, or over time in live cells [33]. Additionally, Ultraviolet Hyperspectral Imaging (UHI) has been used to produce comprehensive molecular imaging signatures [34], demonstrating the ability to distinguish different cellular components based on their independently unique absorption and dispersion properties. Recently, deep UV microscopy has also been applied to hematological analysis for diagnosis of blood disorders [34]. Deep UV excitation is also commonly used by x-ray crystallographers to excite endogenous protein fluorescence, although in that case the imaged light is Stokes' shifted fluorescence emission that is collected at low magnification [35]. Here we report the use of a custom high-resolution deep UV microscope used in a transmitted light configuration for the purpose of enhanced imaging of live *P. falciparum* parasites, and demonstrate clear advantages in performance as compared to visible light. We additionally survey the effectiveness of a standard inverted commercial microscope equipped with near-UV wavelengths as well as a standard visible light trans-illumination white light LED.

## Results

Cultured red blood cells infected with *P. falciparum* were injected into quartz flow cells (S1 Fig) and imaged at multiple wavelengths on a custom-built deep UV microscope (Fig 1, $100 \times /0.85$ glycerol immersion) as well as an inverted commercial microscope using a simple brightfield modality ($40 \times /1.3$ oil immersion). Flow cells were imaged for a maximum of approximately 2–3 hours to avoid parasite health decline outside of incubation conditions. Additionally, deep ultraviolet light exposure was minimized by using a hardware synchronization module that only illuminates the sample for the duration of the camera exposure (S2 Fig). Imaging was performed on freshly-prepared flow cells on multiple distinct dates at various parasitemia levels, and the results were later merged computationally for aggregate analysis.

Our image analysis pipeline used custom data classes in Matlab (Mathworks, Natick, USA) to store, organize, and process RBC images at multiple wavelengths and focal slices (S3 Fig). First, individual datasets were imported for pre-processing. Since the quartz UV objective exhibited substantial aberrations, such as chromatic focal shift and lateral distortion between color channels, focal stacks for each color were re-aligned axially as well as laterally via affine transformation to co-register the three channels. Next, semantic segmentation was performed by a ResNet-50 convolutional neural network (CNN) [36] which we trained by manually segmenting RBCs with diverse appearances and parasite lifecycle stages. Binary masks generated by segmentation were post-processed by an instancing algorithm that separated adjacent cells and also rejected those falling outside a certain size and roundness range, and those intersecting with the image boundary. The primary rationale for morphological filtering was to exclude cells with edge-on orientation, those with high degree of crenation (echinocyte formation—see discussion for more details), and spatially overlapping cells.

The main image classification task—labeling the parasite stage for each RBC (or none if healthy)—was performed using a retrained GoogLeNet CNN [37]. Initial retraining was achieved by manually sorting a ~5,000 count subset of all individual RBC instance thumbnails into specified directories, using the native operating system file explorer (Windows 10 Professional or Mac OS). A machine classifier was trained on this initial subset, which was then able to accelerate the sorting process. Subsequently, larger annotated datasets for training and validation were achieved by exporting a fraction of automatically-classified cells with low confidence scores for human correction ("human-in-loop"), which were used to overwrite the

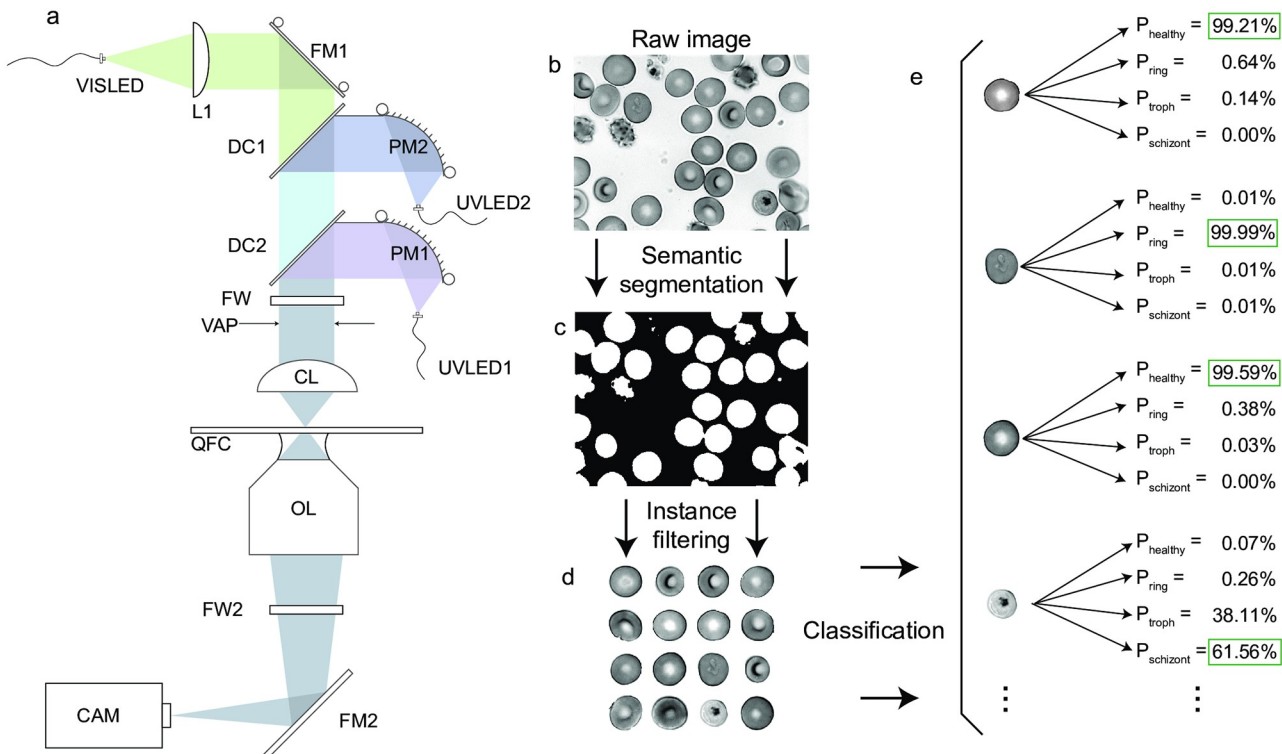

**Fig 1. Overall experiment with a custom UV microscope.** The deep UV microscope (a) was built in a simple transmitted light configuration using a finite conjugate objective lens (OL) to form an image on a camera (CAM) via a rigid fold mirror (FM2) without the need for a specialized tube lens. A custom condenser was built to combine three collimated LEDs (UVLED1, UVLED2, VISLED) using an adjustable fold mirror (FM1) and two dichroic mirrors (DC1, DC2). UV LEDs were collimated using parabolic mirrors (PM1, PM2). Transmitted light numerical aperture was limited with a variable aperture (VAP), then focused onto the sample using a UV fused silica condenser lens (CL). Filter wheels (FW1,2) were added for fluorescence applications (not used in this study). Samples were mounted in Quartz Flow Cells (QFC) for compatibility with deep UV imaging. b) Raw image of parasitized RBCs. c) Binary mask produced by semantic segmentation with a trained ResNet-50 network. d) RBC instances were masked from the raw images and filtered by size and shape parameters to reduce the number of edge-on, misshapen, and/or clipped cells. e) Filtered RBC instances were classified by a retrained GoogLeNet architecture, whose output assigns a probability for each category. Example probabilities are shown for a subset of cells in the raw image.

imperfect machine labels. In this way, we were able to generate large, fully-annotated datasets consisting of 14,219 cells from the UV scope and over 60,000 cells from the commercial microscope (see Table 1 for dataset statistics). Once our datasets were fully human-annotated, we re-trained new classifiers on a random 90% partition of the pooled N best focus slices from all the datasets (N = 5 for UV microscope, N = 1 for commercial microscope). Using additional focus slices served as a natural augmentation of the training dataset size, while simultaneously

**Table 1. Summary of training and validation datasets.**

| Microscope | Objective | Wavelengths (nm) | Healthy | Rings | Trophs | Schizonts | Total |
|---|---|---|---|---|---|---|---|
| UV | 100 × /0.85 | 285 | 12,938 | 734 | 359 | 188 | 14,219 |
| UV | 100 × /0.85 | 285,365,565 | 10,575 | 261 | 124 | 27 | 10,987 |
| Leica DMi8 | 40 × /1.3 | 405 | 56,898 | 5,213 | 2,272 | 583 | 64,966 |
| Leica DMi8 | 40 × /1.3 | 365,405,Broadband | 61,761 | 6,731 | 4,215 | 1,682 | 74,389 |

* UV microscope data additionally included five best focus slices, acting as hardware-based data augmentation system.

** Single- and multi-wavelength datasets consisted of distinct but partially-overlapping merged sets of raw data.

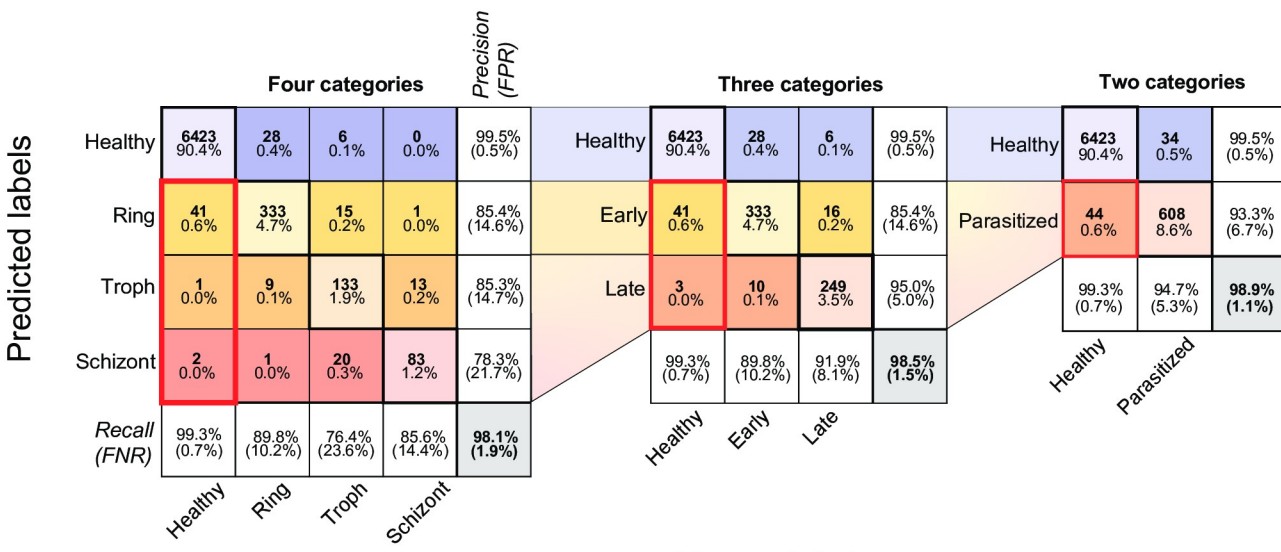

**Fig 2. Classification performance of deep UV microscopy at 285 nm.** Classification was evaluated with either two, three, or four categories. For each level of granularity, confusion matrices are used to represent detailed classifier performance. Diagonal entries correspond to correctly classified instances and off-diagonal entries correspond to incorrectly classified instances. 'Predicted labels' represents the classifier predicted categories and 'Human labels' represents the human-annotated ground truth. Overall classification accuracy is shown in the bottom right cell of each matrix and is defined as the percent of all instances that were classified correctly. The precision for each category is shown along the far right column (false positive rates underneath in parentheses), while the recall is shown along the bottom row (false negative rates underneath in parentheses). Within each cell in the central colored matrix, the total number of counts is shown (bold, top), as well as the corresponding percentage of the total population (bottom). Thick red boxes indicate the rates of healthy cells classified as parasitic. The classifier was trained on a random 90% partition of the pooled dataset containing all five best focus slices, comprising 58,203 healthy, 3,344 ring, 1,566 trophozoite, and 873 schizont images. The validation was performed on the remaining 30% partition consisting of 6,467 healthy, 371 ring, 174 trophozoite, and 97 schizont images.

including examples of slightly de-focused images in the training, in order to reduce the system's dependence on achieving an exact focus.

Using this method, our unmodified four-category classifier was able to achieve a raw overall classification accuracy of 98.1% for custom UV scope images taken at 285 nm. Full confusion matrices are presented in Fig 2, displaying the precision, recall, false positive rates (FPR), false negative rates (FNR), and mis-classification rates for each category. It is important to note that even for the highest expected parasite densities (laboratory or clinical), samples are always composed predominantly of healthy cells, leading to inherently unbalanced data. Such an imbalance places stringent requirements on the FPR of an image classifier, as even low rates of error will result in a large number of healthy cells labeled as parasitic. We also note that unless the culture is artificially synchronized to the late stages, ring-stage parasites typically predominate over the more mature trophozoites, and even more so over the short-lived schizont stage. The highly unbalanced nature of the sample composition biases the cross-entropy loss function and obscures contributions from minority classes during classifier training. In order to compensate for the imbalance during classifier training, we used the following weighted cross-entropy forward loss function:

$$L = -\frac{1}{N} \sum_{n=1}^{N} \sum_{i=1}^{K} \frac{b_i}{n_i} T_{ni} \log(Y_{ni}) \tag{1}$$

Where N is the total number of training images, $n_i$ are the fractional representation of each class, $b_i$ are empirically-determined training biases, $K$ is the number of classes, $Y_{ni}$ are the predictions, and $T_{ni}$ are the targets (human annotated labels). Here we introduced the term $\frac{b_i}{n_i}$ in

order to re-normalize the training weights to account for class imbalance [38]. As a result, training bias towards dominant classes can be eliminated, such that the resulting classifier's FPR and FNR will, on average, be balanced. For our specific training and validation datasets, it was determined that the optimal values for $b_i$ are [4, 2, 1, 1], corresponding to the classes [healthy, ring, trophozoite, schizont], to re-balance confusion matrices that resulted from processing real samples. The relative balance between false positive and false negative rates is further discussed in the context of confidence thresholding and extrinsic validation. Training plots shown in S4 Fig.

## Annular and dendritic ring stage parasites are both observed

Traditional fixation and Giemsa staining procedures modify the sample in a way that improves contrast for transmitted visible light, but may also modify it in other ways [39]. For instance, ring-stage parasites are traditionally identified by their intense staining density around a prominent annular region, thought to be caused by nuclear staining combined with an elevated peripheral density of ribosomes [40]. However, in addition to the canonical annular form, ring stage parasites may also assume a dendritic (or amoeboid) appearance, as observed in both live-cell imaging [13] and electron microscopy [40], and even the dynamic interconversion between the two forms [13]. In the present work we observe both annular and dendritic ring forms with nearly equal regularity in our static images (see Fig 2 and S5 Fig), as well as some limited evidence for dynamic interconversion (see S6 Fig). We note that although the two forms are visually distinct, in this study our classifiers were trained to recognize a spectrum of formations from annular to amoeboid, all as belonging to a singular ring category.

## Post-processing improvements to raw classifier output

In addition to reduction of labor-intensive steps, there are other advantages of machine classifiers over human scoring. The potential for longitudinal reproducibility, elimination of inter- and intra- observer variation, the potential to create a centralized repository with consensus expert annotations, and finally, the ability to analyze numerical classifier confidence scores enables further improvements in performance through statistical analysis. We describe multiple ways in which leveraging these attributes can lead to improved performance over raw classifier output.

**Category merging.**    In Fig 2, we summarize the rates of correct and incorrect classification of all the instances in the validation dataset (a 10% random partition of all the annotated data). The confusion matrices compare the results for all four categories from the raw classifier. However, we noted that in some instances lifecycle stage appears to be transitional, share morphological features common to more than one stage, or simply be difficult to distinguish for other reasons. In particular, rings transitioning to the trophozoite stage began accumulating hemozoin (visible as dark highly-absorbing puncta) with highly variable morphologies, and conversely, some early trophozoites had not yet grown in size but exhibited hemozoin accumulations (see S5 Fig for examples). Likewise, many mature trophozoites had grown large in size and accumulated substantial hemozoin, while some early schizonts had only begun displaying increased cytoplasmic texture indicative of nascent merozoite formation. We therefore found it useful to study the statistics for merged classifiers with three- and two- category schemes, which resulted in higher accuracy by not attempting to distinguish borderline transitional instances, at the cost of decreased granularity. The three-category classifier was created by using a single 'late' category; the summation of the trophozoite and schizont probabilities resulted in higher confidence and higher accuracy. The three-category output better reflected partial information in the case of high total confidence spread across two or more categories.

Our four-category classifier achieves a raw overall accuracy of 98.1%, and combined misclassification of true healthy cells at a rate of 0.7%. We note that by merging the trophozoite and schizont categories, the precision and recall of the resulting 'late' category is substantially improved (Fig 2); as is the overall accuracy (98.5%), reflecting a reduction in the overall number of misclassified cells. Further reduction of the model to two categories (Healthy and Parasitized) results in an overall accuracy of 98.9%, a parasitic recall rate of 94.7%, and a false-positive rate of 0.7%.

**Confidence thresholding.**   We also noted that although the large majority of the population is classified with very high confidence (healthy cell median confidence is 99.9%—Fig 2 and S7 Fig), the confidence distribution has a long tail extending into the low-confidence regime (<90%). This could be explained by many factors including variation of individual parasite anatomy, life cycle stage, relative image focus, non-standard RBC morphology, obstruction by other RBCs, or external stochastic factors such as mechanical vibrations or stage drift. We posited that this subset of cells might exhibit a higher than average error rate, and that excluding it from analysis would be net beneficial to the results. However, it should be noted that removal of data poses an inherent risk of bias and should be applied judiciously. Further, knowledge of the statistics underlying the distribution of confidence scores and their typical correlation with predictive power is essential in applying an appropriate threshold value on classifier confidence. Selection of optimal threshold value is primarily a trade-off between error reduction and introduction of bias. As the threshold value is increased, the rate of rejection of misclassified cells should be higher than the incremental rejection of correctly-classified cells, and the estimate of overall sample composition should improve. However, certain categories may be inherently more difficult to score than others, implying lower confidence values on average. In this case, as the threshold value is increased the more difficult categories will be erroneously rejected at higher rates than easier categories, introducing bias error.

In S7 Fig, we studied the effect of confidence thresholding on our data by plotting the distributions of confidence scores stratified by predicted and human labels, arranged in the form of a confusion matrix. Indeed, the median confidence scores for off-diagonal matrix elements (incorrectly scored cells) were found to be substantially lower than those for on-diagonal positions (correctly scored cells), suggesting that without prior knowledge of ground truth the results could be improved by confidence thresholding. As a performance metric, we considered the estimated overall sample composition accuracy, as opposed to optimization for any one particular element of the confusion matrix. Indeed, the pragmatic output from the machine classifier is the estimated overall sample composition as opposed to the correctness of any one individual cell. Correspondingly, we found the utility of confidence thresholding to be limited in cases where the confusion matrix was already balanced across the diagonal, but more effective in cases where there were more false-positives than false-negatives (or vice-versa). In fact, balanced classifier results were usually worsened with increasing threshold. We further note that the four-category classifier often has difficulty distinguishing late trophozoites from early schizonts, despite a high combined confidence. Confidence thresholding in that case therefore tends to erroneously reject late stage parasites. Merging into a combined 'late' stage category resolves this issue and results in a greater improvement with threshold application.

**Confidence-based post-processing strategies**. The optimal confidence threshold value cannot be determined without prior knowledge of the underlying sample composition. We therefore discuss methods of improving classifier performance in the absence of prior knowledge, and without the need for large-scale annotation efforts.

**False-positive rate parameterization**: As shown in S8 Fig, the median confidence value for the healthy population is typically very high (99.9%), presumably due to the excess of available

healthy cell examples in the training dataset as compared to parasitized examples. Increasing the confidence threshold therefore rejects parasitized cells at a higher relative rate compared to healthy cells, leading to a decrease in the observed parasitemia as a function of threshold value. There is thus a trade-off between elimination of false positives and the recall performance of the classifier. One method of resolving this trade-off is to choose a maximum acceptable false positive rate as an independent parameter, and setting the minimum confidence threshold for which the desired rate is achieved. In S9 Fig, we show the result of analyzing a negative control (healthy) sample with our classifier at 285 nm. As confidence threshold is increased, the FPR is reduced to zero. Recall will be also be modified as a result, but can be compensated for in the resulting measurement of sample composition. For stringent applications at low parasitemia levels, the FPR must be held as low as possible. While recall compensation can offset this known effect, rejection of cells by confidence thesholding correspondingly increases the total number of cell images required to ensure adequate sampling.

**Population sub-sampling**: Human annotation is used to generate labels for a small random sub-sample of the entire dataset. Optimal confidence threshold can then be assessed by minimizing experimental error as a function of threshold for the human-annotated sub-sample, and applying the same value to the remainder of the dataset.

**Human collaboration**: A human annotates a non-random, low-confidence subset of the data, which is most likely to be incorrect. The new human labels are incorporated into the dataset by overwriting the machine labels, and the classification results are updated using the merged data.

**Focus-dependence of classification performance.** In order to ensure that the overall best focus images were captured, we acquired complete focus stacks for all datasets on the custom UV microscope. Since the optimal focus for any one parasite in general does not coincide with the global best focus of the image field of view, any one focal plane likely does not capture optimal views of each cell. Therefore, focal stacks provide an opportunity to improve classifier robustness by capturing a range of potential positions over which parasite features might be used for classification. We studied how various metrics of performance were affected by focus relative to global optimum (see S10 Fig), taking advantage of the additional information to boost performance beyond raw classifier output.

Because relative parasite location and orientation with respect to the global best focus plane was variable and not known *a priori*, we reasoned that robustness could be improved by consideration of all the focal slices available for each RBC instance. Among many possible approaches making use of focal information, our most effective strategy was based on selecting the focus slice with the highest classification confidence for each cell. Each focal slice of each RBC was independently processed by the classifier, but only the slice with the highest classification confidence was used to produce the final label for each given cell. In this manner, different RBCs in the same field of view can be classified independently at their optimal focal plane. We found this method to be more robust across nearly all performance metrics, resulting in a more favorable result than any one focal plane in isolation. When classifying four categories, overall accuracy improved from 98.1% to 98.8%, while the FPR for rings (healthy incorrectly identified as rings) decreased dramatically from 0.7% to 0.2%. This reduction in FPR is especially helpful for estimation of overall sample composition, as the balance against the FNR was nearly equalized. In Fig 3, we quantified several key performance metrics as a function of focal offset for all four categories.

## Wavelength-dependence of classification

The majority of malaria researchers likely do not have the time or resources to build a dedicated deep UV microscope, nor is it practical or economical to fabricate custom quartz flow

                                                     

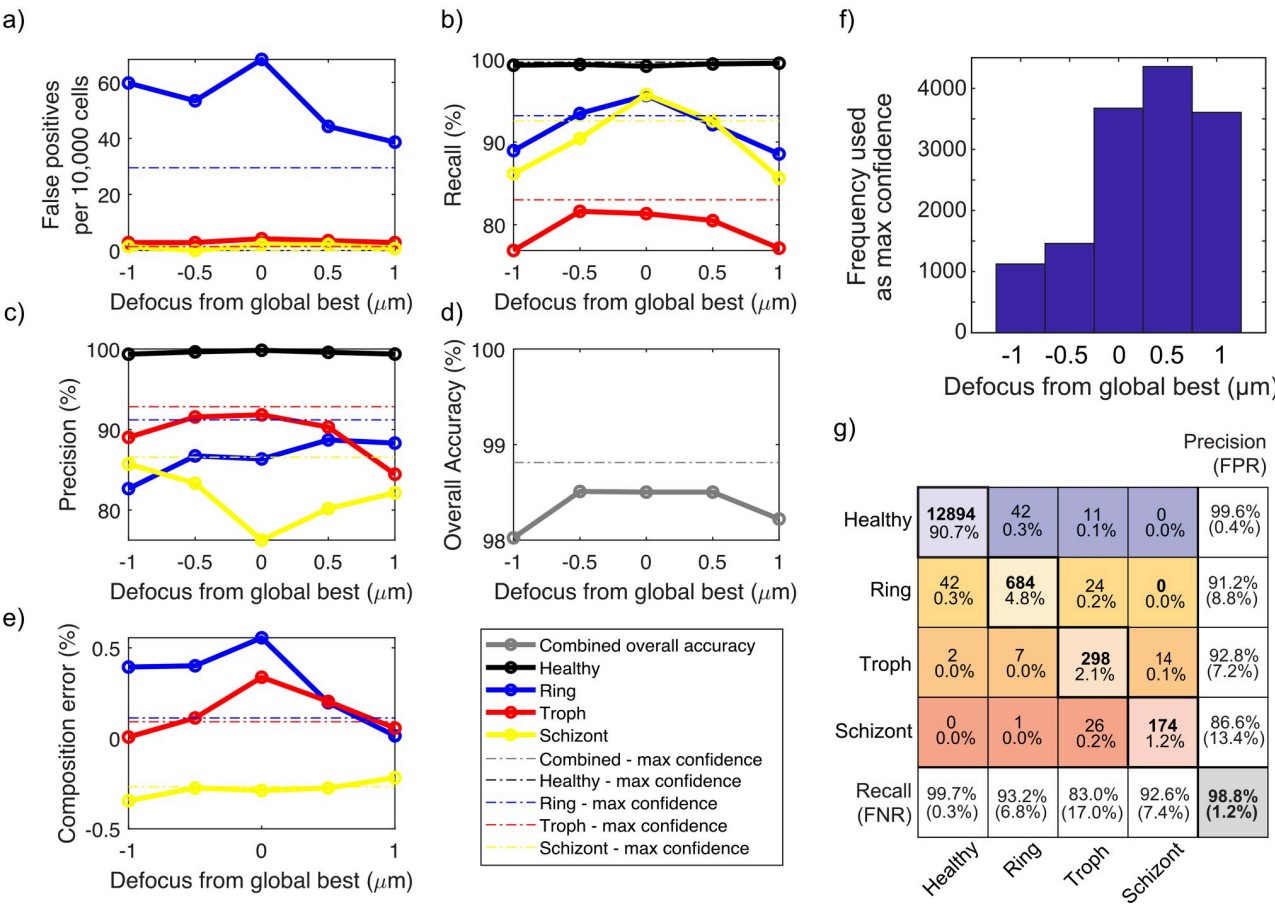

**Fig 3. Analysis of classification statistics vs. focal offset in deep UV images acquired at 285 nm.** In all panels, solid lines refer to statistics from processing all cells at the same global focus offset, while dashed lines correspond to using the optimal focus slice on a cell-by-cell basis, determined via the maximum confidence strategy. a) False positive rate vs. defocus from global optimum for all three parasite stages, b) Recall vs. defocus for all categories, c) Precision vs. defocus, d) Overall accuracy vs. defocus, and e) Sample composition estimation error vs. defocus. f) Histogram showing frequency of slice usage in the max confidence method. Slice number 3 is the global best focus slice, with slices evenly spaced in 0.5 $\mu$ increments. g) Resulting improved confusion matrix after applying the max confidence method. To increase the number of datapoints, all focus-dependent dataset statistics in this figure and its supplementary figures were derived from analyzing pooled classifier training and validation datasets as a function of focus slice.

cells (required for deep-UV imaging) for routine parasite analysis. We therefore explored the possibility of label-free parasite classification at longer wavelengths both on our custom microscope as well as using a commercial light microscope. By producing spatially-registered images at multiple wavelengths, we performed direct comparisons of classifiers across different wavelengths and focal slices, using the same set of human-generated ground truth labels. Refinements to spatial registration were made in post-processing software by selection of the global best focus at each wavelength, and subsequent in-plane affine transformations between wavelengths (see S11 and S12 Figs). Manual annotation was performed only on the highest resolution and global best-focus data, with human-generated labels transferred correspondingly to all other color channels and focus planes. Note that for the wavelength-dependence study, training and validation datasets consisted of distinct, but partially-overlapping sets of cell images, as compared with single wavelength classifier training and validation. For a detailed analysis of resolution, depth of focus, and camera pixel sampling information for both microscopes, see S13 and S14 Figs.

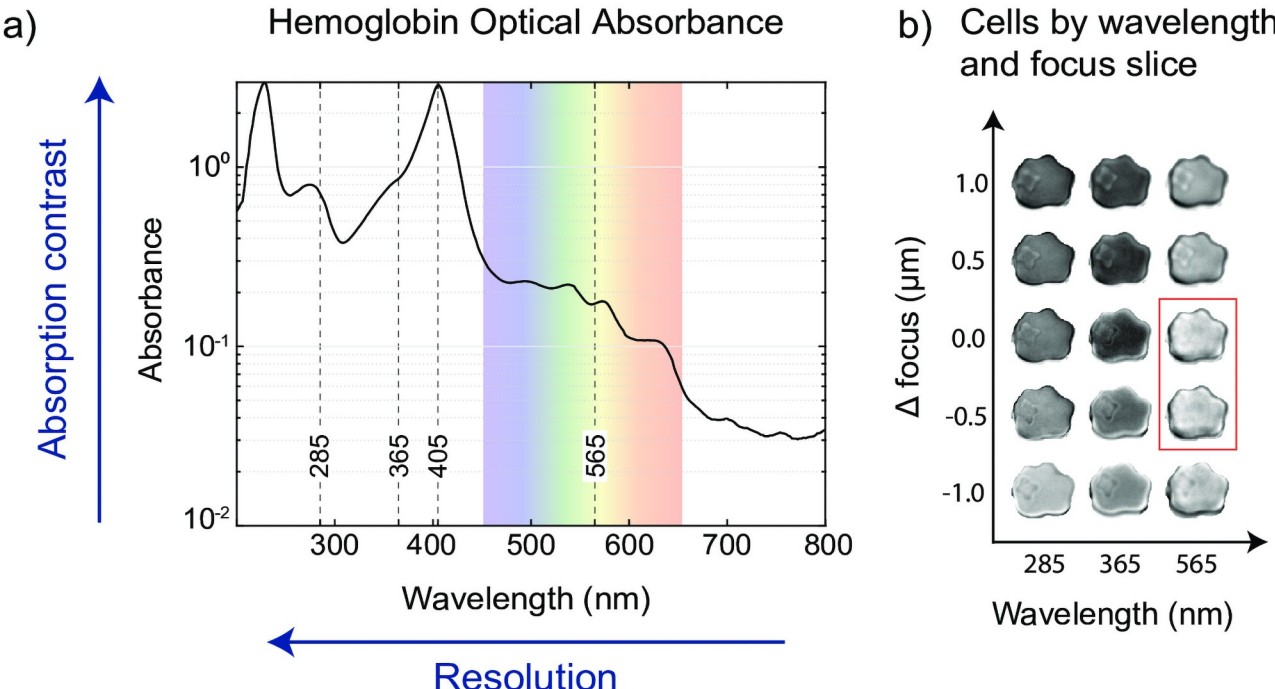

**Fig 4. Wavelength and focus affect parasite contrast and classification efficiency.** a) Optical absorption spectrum of hemoglobin is plotted, with the visible portion of the spectrum highlighted in color. Vertical dashed lines indicate the wavelengths used for imaging on the UV scope (285, 365, and 565 nm), and on the commercial microscope (365, 405, and conventional visible spectrum LED). The hemoglobin (Hb) spectrum is suggestive that parasite identification should be influenced by absorption contrast (highest at peaks), while the shortest wavelengths enable the highest resolution (highest at short wavelengths). b) An array of images of the same ring-stage infected RBC generated by multi-dimensional image registration, enabling assessment of classifier performance as a function of wavelength and focal offset. The parasite becomes nearly impossible to observe in some of the focus slices at 565 nm (red box), illustrating the benefit of increased contrast and resolution.

Observation of RBC appearance over a range of wavelengths, focal planes, and parasite stages permitted qualitative characterization of image features, and quantitative characterization of machine classifier performance. In Fig 4B and 4C, the same RBC is shown across an array of five focal positions and three wavelengths, with additional examples in S15 Fig. It is apparent that the parasites' contrast changes as a function of both variables. This effect can be attributed to the parasites' real and imaginary refractive index components being lower in value as compared to the RBC [14], manifesting as a combination of optical phase and absorption mechanisms, respectively. In particular, parasites can appear as either brighter than the RBC cytoplasm, darker than the cytoplasm, or exactly the same, depending on the focal plane. The phenomenon of through-focus contrast inversion is expected for purely phase objects (no imaginary component) imaged in brightfield [41]; an effect that is also explained by the Transport Of Intensity equation [42]. Early stage parasites without visible hemozoin accumulations in particular exhibit this effect. On the other hand, the imaginary component is directly related to the hemoglobin (Hb) molecular absorbance and contributes to a signal that does not invert sign through focus. The interplay of these two factors can explain the relative prominence of the parasite across the focus-wavelength array. Notably, in Fig 4B, a dendritic ring-form is seen to vanish entirely at certain focal planes when imaged with visible light due to the phase component exactly cancelling the absorptive component of the image on the host RBC backdrop. We remark that ultraviolet wavelengths, possessing both higher resolution and higher molecular absorbance by Hb, are more suitable for robust label-free imaging of *P. falciparum* because a) the parasite membrane is sharply resolved (higher resolution) and b) there are no

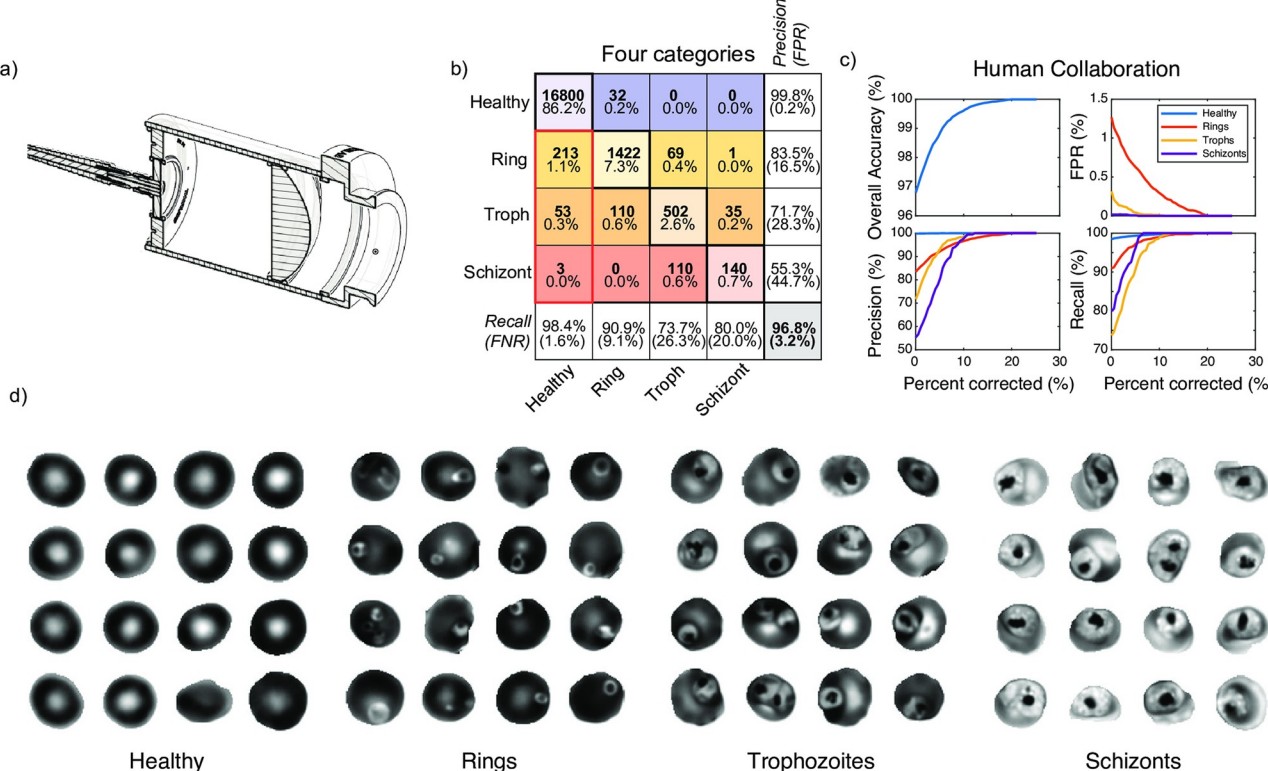

**Fig 5. Commercial microscope with 405 nm excitation.** Equipping a commercial widefield microscope with 405 nm excitation generates high-contrast label-free images of infected RBCs. a) Cross-sectional view of our fiber-coupled, collimated light source used for near-UV excitation through the microscope transillumination port. b) Confusion matrix from the single-wavelength validation dataset of the classifier trained with 405 nm images from the commercial microscope. c) Results from human collaboration, ie. human correction of labels for only the cells with the lowest classifier confidence. All four graphs depict performance metrics as a function of the total percentage of cells corrected. Upper left: Overall accuracy, Upper right: False-Positive Rate, Lower left: Precision, and Lower right: recall. The human plus machine classifier achieves perfect performance after correction of 20% of the dataset. d) Example cell images of all four categories, acquired at 405 nm. From left to right: Healthy, Rings, Trophozoites, and Schizonts.

focal planes for which the cytoplasm fully vanishes by contrast cancellation (absorption contrast exceeds the max phase contrast). We also note that although in many visible light images ring stage parasites were clearly identifiable by phase effect, there were also cases where the definition of the parasite membrane was crucial for human parasite identification, for which the longer wavelengths were not always sufficient to resolve.

**Near-UV modification of a commercial microscope improves classification**. In order to increase the accessibility of our method, we conducted imaging and classification experiments on an inverted commercial microscope using the built-in trans-illumination white LED source as well as a custom light source configurable with two near-UV wavelengths (405 nm, and 365 nm—see Fig 5). Hypothesizing that Hb optical absorbance played a role in parasite detection, we selected these wavelengths according to the spectrum in Fig 4A. In the case of 405 nm, RBC absorption contrast was maximized with respect to both background and parasite cytoplasm, while resolution could theoretically be maximized at 365 nm (the shortest wavelength transmitted by most commercial microscopes), and finally the broadband (white) LED representing the unmodified system. We observed that with the high molecular absorption coefficient at 405 nm, early-stage parasite contrast was apparently dominated by the spatial occlusion of Hb (S6 Fig), causing the parasite images to manifest as bright bodies on dark RBC backgrounds. This follows directly from that fact that early ring forms have not yet consumed a large Hb

fraction, leaving the RBC nearly opaque. Later stages displayed internal enrichment of hemozoin crystal [43], which is also characterized by a spatial re-distribution of pigment throughout the cell. As a result, RBCs infected with later stages of parasite maturation can additionally be distinguished by depletion of pigment from the cytoplasm (Fig 5). Our images acquired at 365 nm were not visibly improved relative to 405 nm, likely due to a combination of optical aberrations limiting resolution (see S13 and S14 Figs for an analysis of microscope resolution, depth of focus, and sampling) and lower absorption by Hb relative to 405 nm.

Incorporation of near-UV wavelengths on the commercial microscope permitted exploration of optimal configurations of our method that are accessible to many groups around the world. We opted to acquire data at lower magnification than the custom microscope, employing a 40 × /1.30 oil immersion objective in order to quickly acquire datasets with large numbers of cells. At 40×, we could acquire approximately 500–1,000 cells per field of view without significant overlapping of cells, increasing statistical resolving power which is important for analysis of low-parasitemia samples [44].

We assessed the performance of the modified commercial microscope by the same metrics as with deep UV, using confusion matrices to compare classifier predictions with human annotations (results from a classifier optimized on a single wavelength dataset at 405 nm is shown in Fig 5, and a direct comparison across wavelengths is shown in S16 Fig). On the basis of per-cell classification accuracy, the results were inferior when compared to the higher magnification and shorter wavelength used in the UV microscope. However, of the three wavelengths, 405 nm displayed the most robust contrast for ring stage parasites, which is corroborated by an exceptionally high recall of 96.6%. The false positive rate was plotted as a function of threshold in S17 Fig on a healthy control sample, showing that on a healthy control, we observe a total FPR of approximately 0.3% at a threshold of 50% confidence, decreasing monotonically with higher threshold. Under ideal conditions, images acquired at 365 nm would presumably have exhibited improved resolution and performance as compared to longer wavelengths. However, our observations suggest that aberrations were a limiting factor (particularly at 365 nm); a trend also reflected in classification performance.

Given the trade-off between raw classification performance and throughput, the most important output from a machine classifier in this context is an accurate estimate of the sample composition, which depends not only on the per-cell accuracy, but also on the balance between FPR and FNR, and the statistical resolving power. The key parameters for both microscopes at multiple wavelengths are summarized in Fig 6. We later assess the sum contribution of these factors by titration of parasitemia, comparing also to the century-old gold standard method of manual inspection of Giemsa-stained smears.

## Direct detection of parasitized RBCs via Faster R-CNN

The above results were used to train and validate a python-based Faster R-CNN model [45] to recognize both healthy and parasitized RBCs directly on raw images, as opposed to performing a segmentation step prior to classification. The method uses a Region Proposal Network (RPN) to first identify image regions likely to contain cells, which are then processed by a unified network to generate object detection confidence scores. Training such a network to recognize malaria presented logistical challenges but had some advantages. First, the direct detection on raw images is simpler to implement, with fewer intermediate steps. Second, the python-based framework is more easily open-sourced and disseminated. Finally, this method has a more direct path for implementation on a low-cost, embedded system.

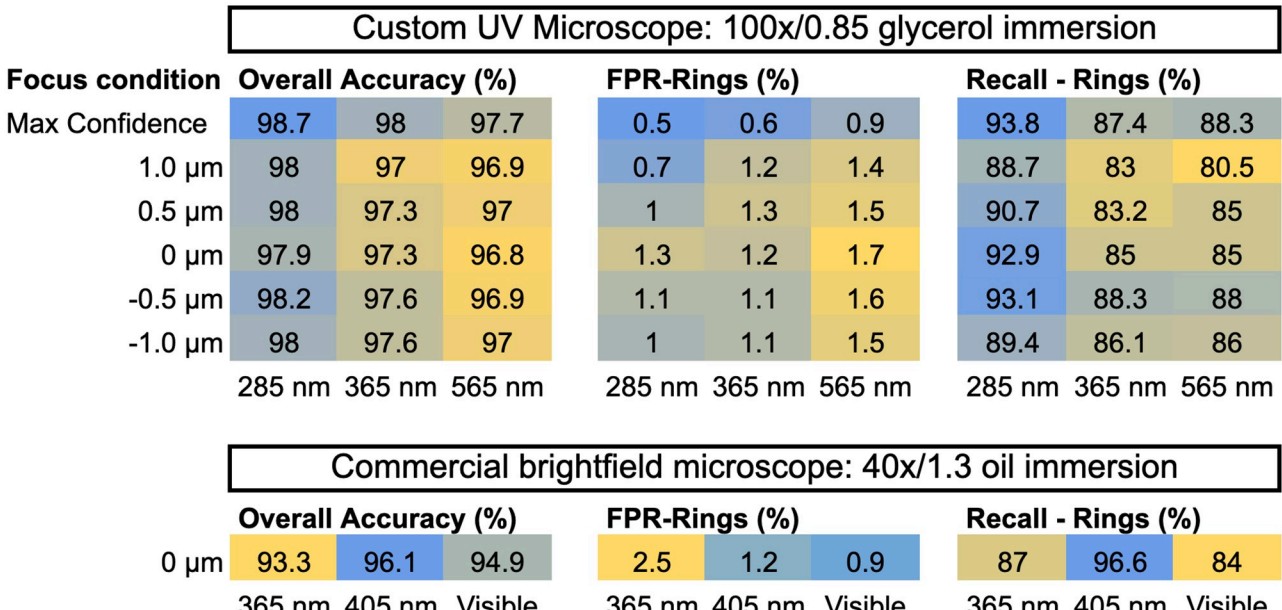

**Custom UV Microscope: 100x/0.85 glycerol immersion**

| Focus condition | Overall Accuracy (%) | | | FPR-Rings (%) | | | Recall - Rings (%) | | |
|---|---|---|---|---|---|---|---|---|---|
| Max Confidence | 98.7 | 98 | 97.7 | 0.5 | 0.6 | 0.9 | 93.8 | 87.4 | 88.3 |
| 1.0 µm | 98 | 97 | 96.9 | 0.7 | 1.2 | 1.4 | 88.7 | 83 | 80.5 |
| 0.5 µm | 98 | 97.3 | 97 | 1 | 1.3 | 1.5 | 90.7 | 83.2 | 85 |
| 0 µm | 97.9 | 97.3 | 96.8 | 1.3 | 1.2 | 1.7 | 92.9 | 85 | 85 |
| -0.5 µm | 98.2 | 97.6 | 96.9 | 1.1 | 1.1 | 1.6 | 93.1 | 88.3 | 88 |
| -1.0 µm | 98 | 97.6 | 97 | 1 | 1.1 | 1.5 | 89.4 | 86.1 | 86 |
| | 285 nm | 365 nm | 565 nm | 285 nm | 365 nm | 565 nm | 285 nm | 365 nm | 565 nm |

**Commercial brightfield microscope: 40x/1.3 oil immersion**

| | Overall Accuracy (%) | | | FPR-Rings (%) | | | Recall - Rings (%) | | |
|---|---|---|---|---|---|---|---|---|---|
| 0 µm | 93.3 | 96.1 | 94.9 | 2.5 | 1.2 | 0.9 | 87 | 96.6 | 84 |
| | 365 nm | 405 nm | Visible | 365 nm | 405 nm | Visible | 365 nm | 405 nm | Visible |

**Fig 6. Wavelength- and slice-dependent classification statistics on the custom UV microscope.** Top: Overall accuracy (left), FPR for rings (center), and recall for rings (right) were selected for detailed breakdown as a function of wavelength (columns), and focal offset (rows). The top row in each table shows the result of processing the data with our maximum confidence strategy for focal plane selection on a cell-by-cell basis. Bottom: The same statistics were plotted for a single focal plane on the commercial light microscope at 40× magnification. For each grouping by microscope and statistic, cells are color-coded on a scale from blue (lowest performance) to yellow (highest performance). All statistics were derived from multi-wavelength datasets, which consisted of distinct but partially-overlapping sets of data with single-wavelength datasets (see Table 1). In both cases, each wavelength-specific classifier was trained and validated uniquely on images at its particular wavelength. To ensure a fair comparison between the wavelengths, all three wavelengths for each microscope shared the exact same random partitions between training and validation, ie. the same distinct sets of RBCs and corresponding ground truth labels were used across wavelengths.

The main logistical challenge to implementing this method was assembling enough human-annotated instances of all the parasite life stages for training. In contrast with pre-segmented RBCs which can be hand-annotated very quickly (see Materials and methods), generation of labels for R-CNN requires carefully drawing and labelling bounding boxes around a large number of cells of each category. Further, the bounding boxes might also contain edges of other nearby cells, adding deleterious background features to the training data. Additionally, since the raw images themselves contain largely healthy cells and very low fractions of trophozoites and schizonts, human annotation performed directly on the raw images would be excessively time-consuming in order to obtain enough annotated examples of rare classes. To resolve this issue, we used our previous image segmentation and classification results to generate class-balanced synthetic raw images for training (total training and validation loss shown in S18 Fig). These balanced synthetic training images for the Faster R-CNN consisted of a random training partition of pre-annotated instances of the various RBC classes randomly distributed over a field of view. Subsequently, the trained network was evaluated on the same series of raw images from which the original training data were derived, but with the analysis statistics drawn only from cells within the validation partition.

The Faster R-CNN method performed well on 285 nm UV images (Fig 7)—better than our original two-step method used for training data generation. Overall accuracy was 99.2%, better than even using slice consensus. Notably, for rings (the most common stage in peripheral blood) the FPR was only 0.2%, with approximately 95% precision and recall. For trophozoites and schizonts the FPR was negligible, and precision and recall near 90%.

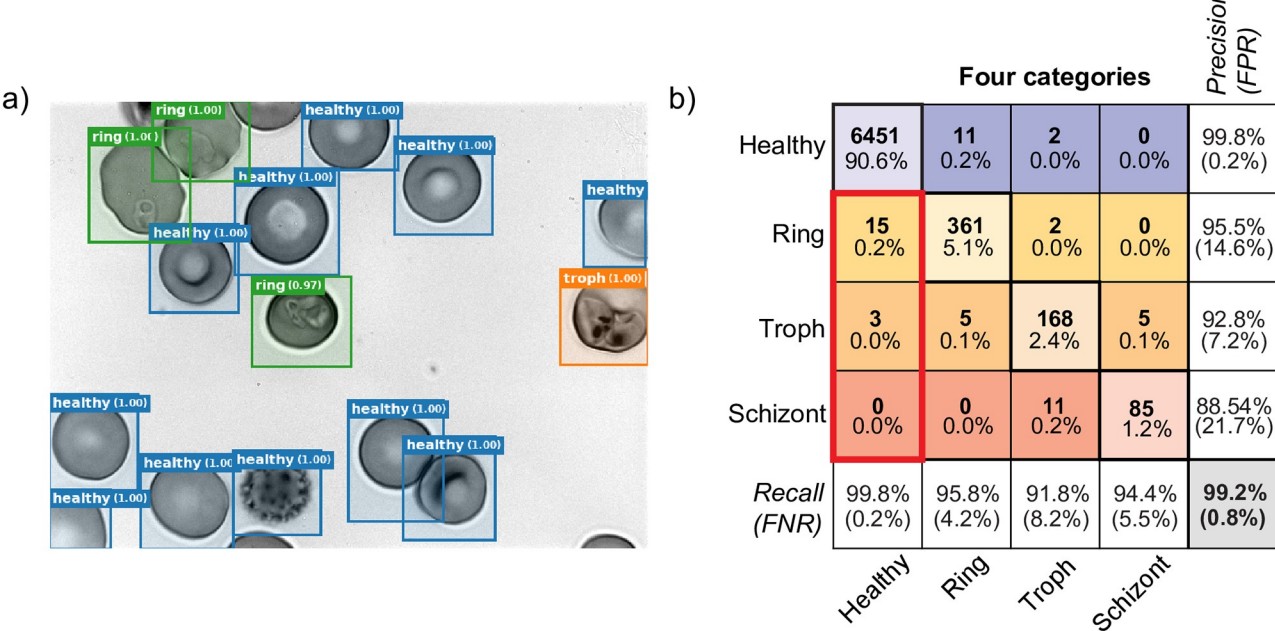

**Fig 7. Faster R-CNN classifier.** a) The Faster-RCNN method detects objects directly on images without using an intermediate semantic segmentation step [45], generating bounding boxes and confidence scores for each detected object. The image shown was acquired at 285 nm, and includes examples of healthy cells, ring stage parasites, trophozoites, as well as an echinocyte (spiky RBC, lower middle-left). b) A four-category confusion matrix is shown for the Faster R-CNN method, using the same format as in Fig 2. Faster R-CNN exhibited a reduced FPR and higher overall accuracy than the two-step method.

## Quantitative extrinsic validation

Thus far we have performed intrinsic statistical analyses using human-annotated images as ground truth, without comparison either to existing methods or with respect to external control parameters. In order to validate our method beyond self-consistency, we conducted a rigorous extrinsic test of our method, by comparing it to hand-counted Giemsa-stained blood smears over parasitemia levels from 18.6% down to less than 0.1%. First, parasites were cultured to high parasitemia, then serially-diluted into fresh, healthy blood at 2% hematocrit, forming a ten-point series. For each point on the curve, the sample was removed from the incubator immediately prior to imaging in order to make a blood smear and load flow cells. Samples were imaged on both microscopes in order to facilitate simultaneous image acquisition, minimizing time delay between the methods. Giemsa-stained blood smears were also prepared at each concentration at the time of imaging, and subsequently counted manually by three experienced technicians. Manual counting was intentionally limited to approximately 300 cells per titration point in order to represent a realistic, routine laboratory counting practice. However, in order to generate a more precise reference point, manual counting was performed on over 2,000 cells at only the highest concentration, by a combination of all three annotators. All nominal parasitemia values were computed based on dilution factors from this reference point.

Results of the comparison are shown in Fig 8. Fundamentally, Poisson counting statistics impose a minimum uncertainty with a variance equal to the mean number of counted parasites. Correspondingly, we expect variance in manual counting of 300 cells to diverge at low parasitemia. To convey this quantitatively, we overlaid grayscale error bands on each plot, each corresponding to one standard deviation of the underlying Poisson distribution for

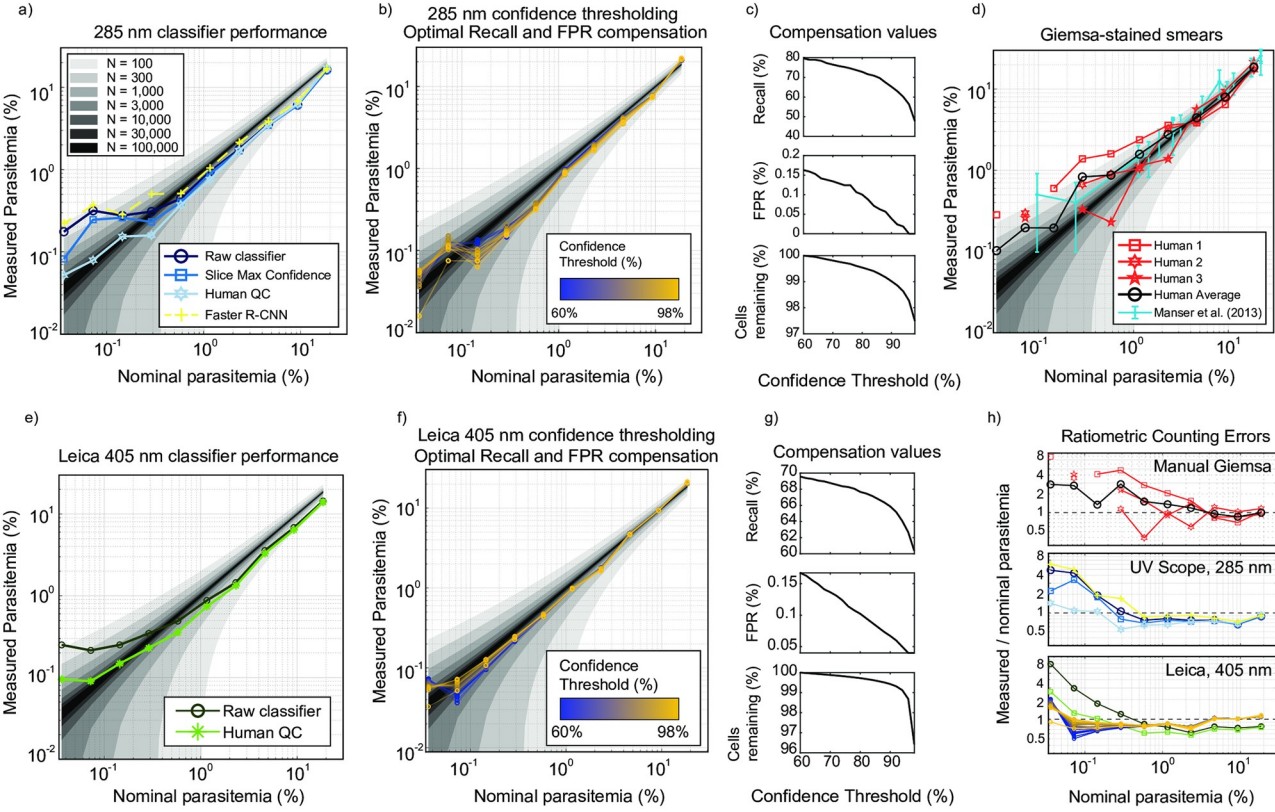

**Fig 8. Parasitemia titration results.** Logscale plots a-b,d-f contain shaded regions corresponding to nominal mean parasitemia plus or minus one standard deviation of the underlying Poisson distribution when counting N total cells, where N is varied from 100 up to 100,000 in roughly 3× increments. The nominal parasitemia in panels a-b,d-f is derived from a manual count of the titration high point using 2,214 total cells, with all remaining nominal diluted values computed by theoretical dilution factor. a) Sample parasitemia extracted from the raw 285 nm classifier results is plotted in dark blue. Post-processing using the maximum slice confidence technique is shown in blue, resulting in a reduction of the false-positive rate evidenced by a further drop in the lowest points of the curve. The light blue data show the result of human collaboration on the putative infected and lowest-confidence healthy cells. The Faster R-CNN method was used to detect cells directly on the raw images (yellow). b) The data from a) were re-plotted after compensation for recall and FPR, for an array of confidence threshold values applied to the data. c) Recall and FPR compensation values are plotted as a function of confidence threshold, which both decrease with threshold value, as healthy cells' median confidence is on the order of 99.9%. d) Manual counting of Giemsa-stained smears by three experienced technicians are plotted with solid red curves (human 1: squares, human 2: hexagrams, human 3: pentagrams), and their arithmetic mean plotted in solid black (circles). Missing markers correspond to points where no parasites were located in the 300 cell sample. Blue data with error bars represents the data from Table 2 in [8], showing the result of extensive validation across hundreds of trained microscopists. Error bars indicate standard deviation across participants' parasitemia estimations. The "Reference mean parasitemia" column from the cited table was used as x-values during overlay. e) The titration results were plotted for the classifier trained on commercial microscope at 405 nm. Dark green circles: raw classifier results, Green '+' signs: Compensated classifier results, and light green '*': after human collaboration, as described in a). f) Commercial microscope classifier results after compensation, for various confidence threshold values. Large cell count numbers permit low relative counting variance, reducing deviation from the underlying distribution. g) Compensation values for recall (top), FPR (middle), and remaining cells after thresholding (bottom) from the data in f). h) Ratiometric error of the data in panels a,b,d,e, and f) vs. nominal parasitemia. Top: Manual counting of Giemsa-stained smears (d), middle: UV Scope classifier at 285 nm (a,b), and bottom: Commercial microscope images at 405 nm (e,f). All data in h) are referenced to the legends in their corresponding raw data panels. All titration data was purely 'test data', acquired after classifier training and validation had been completed.

several values of total counted cells, demonstrating how resolving power improves as a larger number of cells is counted. All bands are relatively narrow at high parasitemia values, and begin expanding asymptotically as they approach an expectation value of one parasite per sample—below which the technician (or algorithm) is not likely to encounter any parasites in the sample.

Our results in Fig 8A demonstrate that with deep-UV microscopy, our raw classifier exhibits a linear response from the high point of 18.6% parasitemia down to approximately 0.5%,

where it is further limited by the detection of false positives. By capturing many thousands of cells per titration point (at 100× with z-stacks), the dataset exhibits superior counting statistics than manual counting of 300 cells—albeit with reduced recall– as defined by comparison with manual inspection of Giemsa-stained smears. The FPR can be modestly reduced by applying the slice max confidence technique (blue curve in Fig 8A), and substantially improved by human collaboration (light blue in Fig 8A). In the latter case, we exported a pool of all 1,595 putative infected cell images as well as the 5,000 lowest confidence putative healthy cells for manual inspection. Image export and label correction was performed on the pool, without knowledge of the underlying titration point for any given RBC. In total, a net 124 cells labeled as parasitic by the classifier were re-labeled as healthy during collaboration, out of a total of 75,343 cells in the dataset. The procedure was performed on a pool of ten total imaging experiments and consumed 90 minutes of hands-on time. It can therefore stand to reason that a single experiment could be human-corrected in approximately ten minutes. Aside from consistently lower parasitemia estimates, our human-collaboration results exhibit superior ratiometric error as compared to both manual scoring of Giemsa slides and raw classifier output. Faster R-CNN analysis of the same data results in very similar performance as the raw classifier, but with marginally higher recall and FPR.

Although human collaboration increases classifier performance for the 285 nm data at 100×, we note that to the extent that the error statistics are stationary, both recall and FPR can be compensated for by introducing multiplicative and additive correction factors to the sample composition estimates. In Fig 8B, corrections were made to the raw data such that each compensated point $C_{comp} = (C_{raw} - FPR)/recall$, and plotted for a range of confidence threshold values. Compensation values were co-optimized by weighted least squares fitting to the ten-point dilution series, whose fit values are shown as a function of confidence threshold in Fig 8C. These data suggest that the classifiers' inherent false-positive rate as measured by best fit to an extrinsic control variable (parasite dilution factor) is significantly lower than the values reported in Fig 2. However, it is important to note that all reported statistics depend to some extent on characteristics of the sample. Specifically, deviations from the discocyte cell morphology tend to degrade classification performance, as evidenced by a high fraction of low-confidence and misclassified cells consisting of echinocytes and spatially-overlapping cells. Notably, the fresh RBCs used in the titration displayed a particularly low number of echinocytes. We hypothesize this to underlie the lower observed FPR in the titration as compared to intrinsic validation. To further investigate this effect, we retrospectively hand-annotated a test partition of the titration dataset in order to assess raw classification performance with low echinocyte fraction (S19 Fig). Indeed, the FPR decreased dramatically to 0.1% for rings, and the recall for healthy cells up to 99.8%.

Titration data collected on the commercial microscope using 40× magnification at 405 nm facilitated acquisition of a far greater number of cells per condition, by virtue of a larger field of view and single focal plane acquisition. We captured images of 20,000–130,000 cells per condition—an order of magnitude more data than with deep UV, leading to a further improvement in counting statistics. Despite inferior intrinsic validation performance, we observed that imaging of a large number of cells led to superior results. The raw classifier curve in Fig 8E is more amenable to compensation, with the majority of datapoints lying within one standard deviation for the N = 30,000 band and many points lying within the N = 100,000 band down to 0.07% parasitemia. We also observe from Fig 8F that although confidence thresholding aids in reducing FPR and therefore is beneficial at low parasitemia, there is only minor additional benefit beyond data compensation. Fig 8C and 8G exhibit similar trends, showing that FPR reduction occurs with increasing threshold, at the cost of lost recall.

## Discussion

This work demonstrates that visible and ultraviolet brightfield microscopy images of live *P. falciparum*-infected red blood cells can be classified by retraining existing deep-learning networks. We validated this method intrinsically by statistical analysis of confusion matrices, and extrinsically by titration of the parasitemia alongside manual scoring of Giemsa-stained smears. Our data suggest that the method exceeds the performance requirements for routine laboratory analysis of parasite cultures from below 1% up to at least 18% parasitemia [46], and when compensated for known recall and FPR rates, likely exhibits a limit of detection lower than 0.1%. By virtue of improved sampling power, automated imaging and classification offers superior performance over manual inspection of thin smears for laboratory samples, while eliminating the time-consuming and variable steps of fixation, staining, and manual inspection.

We employed a custom-built deep UV microscope equipped with wavelengths as short as 285 nm, as well as a modified commercial light microscope operating from near-UV to visible light, in order to explore performance across a range of imaging conditions. Our results suggest that image classification performance depends on both image resolution and contrast: on our custom microscope, the best intrinsic classification was observed at 285 nm / 100× where resolution was maximal. Of the wavelengths accessible to commercial microscopes, 405 nm performed best. While resolution theoretically improves with lower wavelength, our commercial system was in practice limited by aberrations and pixel sampling considerations; it is therefore reasonable to expect that addressing these issues could further improve results in that regime. In conjunction with resolving power, contrast is the result of light-matter interactions involving both the real and imaginary refractive indices of the parasite and RBC cytoplasm [41, 42]. We found that the phase effect (real component) is sensitive to and inverts in sign through image focus, but the absorption effect (imaginary component) results in positive parasite contrast independently of focus. Over the wavelengths we tested, Hb absorbance varied by approximately a factor of 15 (see Fig 4A), and in the case of 405 nm appears to be large enough such that ring parasites are visible in stark positive contrast with respect to the RBC cytoplasm, as compared with other wavelengths where it is possible in certain focal planes for ring stage parasites to entirely vanish (Fig 4B). We hypothesize that this factor explains the high recall rate for rings at this particular wavelength. Independent of wavelength, the use of information from multiple focal planes consistently increases the likelihood of identifying parasites (or ruling them out).

Our results using a standard commercial microscope extend the impact of this work beyond the limited set of researchers with the time and resources to build a custom deep UV microscope. The simple addition of a 405 nm light source enables a boost in performance, effectively increasing the robustness of information used by the machine classifier. We have made all of our software open-sourced, including resources for rapidly generating large annotated datasets for re-training on other microscopes. We anticipate that adopters of this technique can quickly replicate this work on existing microscopes by first using our published, pre-trained classifiers to sort RBCs imperfectly, then provide corrections to the labels by careful inspection of a subset of low-confidence cells, and then using the resulting dataset to train an improved classifier. In our experience, this procedure requires less hands-on time than generating training labels from scratch, especially because as the classifier improves, the fraction of cells requiring re-labelling diminishes and median machine confidence increases. Since our software keeps a record of all the existing annotated files across iterations, repeated annotations of the same cells does not need to be performed across iterations.

Titration of parasitemia over a wide range suggests automated counting exhibits, by most metrics, better performance than manual counting of Giemsa-stained thin smears, which is the gold standard in the field. With the compensated commercial microscope datapoints in Fig 8F lying close to the expected single standard deviation band for 100,000 counted cells, it is possible that the limit of detection had not yet been reached even at the lowest point on the curve (0.036% nominal parasitemia). Similarly, ratiometric errors shown in Fig 8H indicate little divergence from unity for human collaboration (middle, 285 nm) as well as compensated data (bottom, 405 nm). On the other hand, manual scoring of blood smears exhibits larger variation over most of the experimental range due to the effects of limited sample size, noting that averaging results from all three annotators substantially reduced error. Finally, it should be noted that although all nominal parasitemia data was generated from a "deep" manual count, the three annotators nonetheless separately assessed substantially different results of 17.4% (N = 1,332 cells), 19.1% (N = 460 cells), and 21.6% (N = 422 cells), which presumably was due both to Poisson sampling error and variation in annotation accuracy between humans. From this we conclude that there is substantial uncertainty in the standard itself to which we reference, highlighting the very need for more consistent and robust counting methods. In fact, according to World Health Organization guidelines [9], malaria microscopists are graded on a competence scale of 1–4, where the highest competence level requires a parasite counting accuracy within 25% of the correct answer only 50% of the time—our three annotators' scores lie well within this window from their aggregate mean.

This work lays the foundation for future developments, such as label-free clinical diagnostic modules for use in low-resource settings. In particular, brightfield imaging requires no fluorescence excitation or emission filters, reducing cost and complexity compared to other automated scanning methods [28]. Further, liquid samples might be processed under flow conditions, avoiding the need for expensive motion-scanning stages. Low-cost computation can be readily achieved in the context of an increasing number of sub-$100 embedded modules specialized in deep model inferencing (examples include Google Coral and Nvidia Jetson Nano), fully-capable of executing classification models on embedded systems at high speeds. However, such a system would likely encounter several challenges. First, clinical parasitemia levels vary widely [44, 47], and must be distinguished from healthy with a high degree of confidence. As a result, clinical diagnostic procedures typically include both thick and thin blood smears. Thick blood smear sensitivity also varies widely between reports [44]. A comprehensive statistical analysis of the literature [44] quotes a median probability of 29.7% that trained manual microscopy technicians will detect parasite levels of $100/\mu L$, and similarly, can detect $1000/\mu L$ with 60% probability—the latter being approximately equivalent to 0.02% parasitemia, or slightly below the lowest dilution point in our titration (0.036%). Future extensions of this method will therefore require increased imaging throughput to match the sensitivity of clinical thick blood smears. Second, individuals may be infected with more than one *Plasmodium* species (we did not perform speciation in this work), complicating the classification task. Third, screening whole blood will present additional challenges such as the presence of lymphocytes, platelets, and variable patient RBC counts. It is also worth noting that in many of our datasets, we observe a relatively large fraction of echinocytes, or RBCs exhibiting spiked morphologies [39]. Previous work has shown that echinocytes and other exotic RBC morphologies can be influenced by the confinement between glass surfaces, and can also depend sensitively on the details of sample preparation [48]. Our flow cells fall within the regimes reported in the literature as enhancing this effect, which we observed with regularity in our experiments. Fortunately, our classifiers could in general distinguish the echinocytes' sharp spicular morphologies from parasites, albeit at higher error rates. Depending on collection methods and imaging consumable design, clinical diagnostic imaging in whole blood may or may not suffer from this effect.

## Conclusion

Implementation of label-free imaging and classification of live *P. falciparum* will enable major reductions in technician time, training, reagent cost, and observer variability with respect to the analysis of parasitized blood samples around the world, in both laboratories and ultimately in the field. Our results strongly suggest that future advances in label-free imaging could enable rapid, automated, and highly accurate clinical diagnosis of fresh, whole blood—replacing error-prone and labor-intensive manual slide preparation and counting, a process that has changed little in over a century.

## Materials and methods

### Ethics statement

Blood for this study was drawn with approval under the UCSF IRB 10–02381. All donors provided written consent.

### Hardware

**Deep UV microscope.** Deep UV microscopy was performed on a custom-built microscope (Fig 1) using a 100 × /0.85 Ultrafluar quartz objective lens (Zeiss, Oberkochen, Germany) to form images onto a UV-sensitive camera (PCO.Ultraviolet, Kelheim, Germany). A multi-wavelength condenser was built for transmitted light excitation using commercially-available UV LEDs to illuminate the sample with 285 nm (Thorlabs #M285L5, Newton, USA), 365 nm (Thorlabs #M365FP1), or 565 nm (Thorlabs #M565L3) light. The condenser used one dichroic mirror to combine the two UV wavelengths, and a second dichroic mirror to merge visible light from the third LED, or alternatively, any standard microscopy source. Each UV LED was collimated with an off-axis parabolic UV-enhanced aluminum mirror (Thorlabs #MPD129-F01) before being combined. An adjustable iris set fully open to 12 mm diameter limited the numerical aperture of the illumination to 0.23.

**Commercial microscope.** All commercial microscope experiments were performed on a DMi8 (Leica Microsystems, Buffalo Grove, USA) equipped with a Plan Apochromatic 40 × /1.30 oil immersion objective (Leica #11506358), motorized XY and focus stages, and custom automated image acquisition software. To facilitate near-UV wavelengths, we constructed a fiber-coupled equivalent of a commercial pre-built microscopy light source (Thorlabs #M405L3-C2) and attached it to the TL-port of the microscope. Our custom source used a multimode fiber which could be moved between multiple LED sources. The microscope's standard broadband trans-illumination LED was used for visible light experiments as a comparison. Resolution, depth of focus, and additional technical detail on both microscopes is provided in S13 Fig.

### Software

**Instrument control.** The deep UV microscope was controlled using custom classes, functions, and scripts written in MATLAB. All software is freely available here: https://github.com/czbiohub/UVScope-control. Overall architecture is detailed in Fig 1 and S22 Fig. The microscope operating system is implemented by a custom "UVScope" data class, orchestrating all the high level processes such as state of the system and interaction with each of the hardware device classes. Other data classes act as hardware interface drivers, image data processing, metadata generation, and storage.

**Image processing.** Two different image processing pipelines were used in this work. The main results (Figs 1–6) all used a two-step method employing semantic image segmentation of

all the RBCs, which were subsequently classified by a network trained to distinguish all four categories of cell. The two-step method is detailed in S3 Fig. We also employed a single-step method (Faster R-CNN).

**Two-step method (segment and classify)**. All two-step method training and classification was performed using the Matlab Deep Learning Toolbox and custom data classes to store and organize the image data, and deep network training was performed on a Windows 10 desktop PC with 128 GB of RAM, and an Nvidia GeForce GTX1060 6144MB GPU card (NVidia, Santa Clara, USA). Image datasets containing multiple fields of view, wavelengths and focus slices were imported, registered, segmented, annotated, trained, and validated (S11 Fig). Since multiple distinct datasets were required in order to increase the size of the training dataset, they were merged prior to network training. After dataset merging, the various other dimensions of the data (fields of view, wavelengths, focus slices, dataset IDs) could then either be merged, kept separate, or both, for purposes of classifier training and validation.

Manual human annotation was performed to provide ground truth training data. In order to generate enough annotations, cropped RBC image instances were written to disk and sorted into labeled directories. Using this method, cells could be manually labeled at a rate up to several thousand cells per hour, which was significantly faster than other methods we explored. A human-in-loop strategy was implemented to further increase the quantity of training data, whereby a network was initially trained on a small dataset ($\sim 5,000$ cells), then used to classify a larger dataset. The 5,000 cells with the lowest confidence scores were exported for human annotation and re-imported for iterative training. In all cases, data augmentation was used during network training and included rotation, scaling, translation, and reflection. All software used for the two-step method can be found here: https://github.com/czbiohub/Label-Free-Malaria.

**Single-step method (Faster R-CNN)**. For Faster R-CNN analysis we modified the Luminoth package made available by Tryolabs [49]. For training, comma-separated-value (.csv) files (produced by our two-step method) containing the locations and labels for each cell were imported. The initial dataset was randomly split into separate training (90% of data) and validation (10% of data) datasets. The datasets were then converted to tensorflow records for training. Hyperparameter tuning was used to optimize validation loss function after iterative network training. The hyperparameters tuned include: the loss function for RPN and R-CNN layers in Faster R-CNN, weights of the RPN and R-CNN loss functions, data augmentation techniques (image scaling, reflection, and rotation), and learning rate. For training, simulated raw images were constructed to enrich the density of low-frequency parasite stages, and were constructed by assigning random locations and orientations for each pre-segmented RBC from the two-step pipeline. All of our codebase is publicly available at: https://github.com/czbiohub/luminoth-uv-imaging/.

**Processing commercial microscope images**. Images from the commercial microscope did not require slice selection. However, alignment between color channels was performed using 2D cross-correlation (rigid body translation). Subsequently, all steps were identical to UV scope image processing.

## Sample preparation

**Cell culture and manual blood smear counting.** *P.falciparum* strains W2 and 3D7 were cultured in 50 mL flasks containing RPMI (Thermo Fisher #31800089, Waltham, USA) supplemented with 0.5% Albumax II (GIBCO 290, Life Technologies, Carlsbad, USA), 2 g/L sodium bicarbonate, 0.1 mM hypoxanthine, 25 mM HEPES (pH 7.4), and 50 μg/L gentamicin. Cultures were maintained at 2% hematocrit in a temperature and gas-controlled environment set to 37˚C, 5% oxygen, and 5% $CO_2$. Cultures were periodically split in order to maintain

parasitemia levels of 1–5% in order to prevent overgrowth, with daily media changes. They were checked daily by briefly centrifuging 500 μL of culture, aspirating the supernatant, and adding 10 μL of the infected red blood cells to a clear glass slide. To visualize *P. falciparum*, standard Giemsa staining technique was used and the slide was subsequently viewed using a Zeiss microscope at 100× magnification.

**Preparation for microscopy.**   Prior to imaging, cultures are diluted two-fold into culture media in order to reduce the likelihood of RBCs overlapping during imaging. 60 μL of diluted cell culture is loaded directly into a custom quartz flow cell (S1 Fig). Flow cell ports are subsequently sealed with clear nail polish, and all outer surfaces cleaned with isopropanol. Quartz flow cells were fastened into a custom 3D-printed adapter with exterior dimensions of a standard SBS well plate. The adapter is directly mounted on the microscope stage.

**Enrichment of schizont stage.**   Classifier training requires numerous examples of each category. Since schizonts were the shortest-lived and least frequently-observed life cycle stage, we synchronized the life cycle stages four days prior to imaging by adding a 5% sorbitol solution to ring-stage parasites for 10 minutes, ensuring that all trophozoite or schizont-infected RBCs were lysed. Synchronization was confirmed in the following days by regular smearing of the parasite culture, and imaging took place when schizonts were abundant.

**Titration of parasitized red blood cells.**   Malaria was grown to high parasitemia by splitting a ring-dominant culture to 5% parasitemia into freshly-drawn, healthy RBCs two days prior to the experiment. Frequent media changes were made subsequently, but no further splits were done prior to the experiment. A ten-point serial dilution into healthy RBCs was performed in 2× increments. The entire series was prepared into flasks in the morning, and stored in the incubator at 37°C prior to imaging. Each sample was taken out of the incubator and loaded into a flow cell on demand, starting with the lowest concentration. Dilutions were made using RPMI complete medium at a hematocrit of 2%, with the healthy RBCs stored in the incubator at 2% hematocrit after the blood draw. As with all other conditions, the sample was further diluted twofold in PBS in order to prevent the overlap of RBCs while imaging. Each concentration point was imaged immediately after loading the flow cell. Giemsa-stained blood smears were performed at each concentration and counted manually as described above.

## Image annotation and ground truth generation

Ground truth labels were generated by manually sorting exported RBC images into labelled directories, by using a standard Windows 10 explorer window to display the images as large icons. Incorrectly labelled cells were moved by a drag and drop to their new directory. In the case of UV microscope, images from the 285 nm channel were exported for sorting and labels applied to both other wavelengths, whereas in the case of the commercial microscope, 405 nm was used for sorting. In the latter case the image contrast and dynamic range were too high for viewing on the monitor as a result of the very high Hb absorption coefficient. To circumvent this issue we computed the logarithm of the image pixels using FIJI (Fiji Is Just ImageJ) after the original export operation. Once the images were sorted, their filenames were used to apply the new labels to the original (unprocessed) images.

**Specific human annotation criteria.**   Manual sorting was used to generate ground truth labels to the following standards (for example labelled parasite images see Fig 2 and S5 Fig). The following paragraphs outline class- and microscope-specific annotation criteria. In general, cells without any of the following features were labelled as 'healthy', regardless of other morphological abnormalities, debris, or otherwise indistinguishable features.

**Deep UV—ring stage**. Ring stage parasites were primarily identified by a membrane boundary contour defining the parasite morphology, which varied greatly but could be

identified either by dendritic projections, canonical annular morphologies, or more rarely, circular blob forms. Importantly, the parasite contrast with respect to the background Hb could be either positive or negative depending on the relative position of the parasite with respect to the focal plane. This effect can be seen in the z-stacks shown in Fig 4.

**Deep UV—trophozoite stage**. Trophozoite stages were distinguished from rings by their moderate size, increased shape solidity (lack of projections or annular voids), and varying stages of hemozoin crystal accumulation. Trophozoites were distinguished from schizonts by their lack of nascent merozoites and their less centralized hemozoin accumulation. RBCs infected by mature trophozoites exhibited brighter cytoplasmic pixel background values due to the parasite's Hb sequestration effectively lowering the cytoplasmic absorption.

**Deep UV—schizont stage**. Schizont stage parasites were identified by their dominant size (usually greater than half the cell area), prominent and centralized accumulation of hemozoin, and their unique characteristic texture due to merozoite formation. However, since early stage schizonts shared many features with mature trophozoites, the distinction was sometimes subtle or non-existent.

**Near UV—ring stage**. At 405 nm, the Hb absorption coefficient is nearly threefold higher than at 285 nm (see Fig 4 for a full spectrum) and the resolution was lower. The effect of high absorption dominated the image contrast, enabling ring-stage parasites to be identified by the parasites' exclusion of Hb. We could identify ring stage parasites in 405 nm images based entirely on this effect, combined with the characteristic range of shapes they are known to assume, based partially on our experience with the higher resolution deep UV images acquired at 285 nm.

**Near UV—trophozoite stage**. Since the Hb absorption was so high, we could easily identify puncta of accumulated hemozoin within the parasite, which, along with increased size and solidity, was the primary means of distinguishing trophozoites from rings. Additionally, the cytoplasmic depletion of Hb becomes more pronounced with mature trophozoites.

**Near UV—Schizont stage**. At near UV our resolution was not sufficient to resolve merozoite texture to the degree that was possible with deep UV, which hindered our ability to distinguish schizonts from mature trophozoites. Overall size, centralized hemozoin accumulation, and RBC Hb depletion were the main image features used to score schizonts at near UV.

## Hemoglobin absorption measurement

Lyophilized hemoglobin was purchased from Sigma-Aldrich (#H7379–1G) and dissolved into Dulbecco's Phosphate Buffered Saline (DPBS) at 10 mg/mL. Absorption measurements were performed on a SpectraMax M3 plate and cuvette reader (Molecular Devices, San Jose, USA) using a 1 cm path length. UV-transparent cuvettes were used (Thermo Fisher #13–878-123) to provide adequate transparency over the range of the measurement.

## Supporting information

**S1 Fig. Flow cells were constructed as follows.** 1 mm circular holes (shown in red below) were laser-cut into quartz slides (Ted Pella 26011, Redding, USA) outlined in blue, using a ULS-P150D laser-cutting system (Universal Laser Systems, Scottsdale, USA). Gaskets (green) were laser-cut from Parafilm (Sigma Aldrich P7793, St. Louis, USA). After cutting, gaskets were aligned by hand, then pressed down gently around the exterior. Quartz coverslips (Ted Pella 26014) were aligned by hand to the gaskets, and then pressed gently. The assembly was sealed by compression of the flow cell (protected inside a single layer aluminum foil envelope) with a mass of 300 g for five minutes, on a hot plate set to 75°C.
(TIF)

**S2 Fig. Design of the hardware sync module.** a) Schematic of the simple circuit used to synchronize LED emission with camera exposure. The circuit uses an analog switch to connect either a programmable analog voltage signal (AIN) or ground (GND) to the output, gated by the camera exposure's logic signal. In series with the analog switch is a manual toggle switch to optional bypass the analog switch, disabling the sync module. b) PCB layout diagram for the hardware sync module. Note that coaxial connector footprints were replaced with standard 0.1" headers, and connected via wires to panel-mount BNC connectors. The circuit was fabricated as a PCB using an LPKF Protomat S103 circuit mill (LPKF, Garbsen, Germany). c) Solid model screenshot of the 3D-printed enclosure for the hardware sync module.
(TIF)

**S3 Fig. Single-wavelength image processing pipeline.**
(TIF)

**S4 Fig. Statistics from training the 285 nm classifier.** The upper plot shows the overall accuracy as a function of iteration and epoch number. The blue line plot (no markers) shows training dataset accuracy while the black line plot (dashed with circle markers) shows validation accuracy. The lower plot shows the loss function (red with no markers: training loss; black with circle markers: validation loss.
(TIF)

**S5 Fig. Example RBC instances from each category in the confusion matrix for the UV Scope at 285 nm wavelength.** The arrangement of the categories is the same as Fig 2 from the main text. For this montage, a maximum array size of 10 x 10 was used for display. Note that for many of the categories (especially off-diagonal entries) there were not enough examples to populate the array, in which case the space was left blank. All data from this figure was taken from the validation dataset, as opposed to the training dataset.
(TIF)

**S6 Fig. Example RBC images acquired on our commercial microscope for all three wavelengths.** (left columns: 365 nm, middle columns: 405 nm, right columns: standard visible LED). Four example cells are displayed for each category, where each row within a category are images of the same physical cell. Ring stage parasites are seen to be highly mobile, as evidenced by their differing positions and even morphology between acquisitions at different wavelengths, which were separated in time on the order of a few minutes.
(TIF)

**S7 Fig. Classifier confidence statistics indicate mis-classified cells are more likely to be low-confidence.** In all three plots, log-scaled histograms of classifier confidence scores are displayed in the pattern of a confusion matrix. Statistics are derived by evaluating the validation dataset from the 285 nm classifier. Total instance counts for each matrix element as well as the medians of the distributions are shown as text insets.
(TIF)

**S8 Fig. Confusion matrices from the custom UV scope at 285 nm show improved statistics after rejection of RBCs based on meeting the minimum classifier confidence criteria.** These confusion matrices show the result of applying the empirically-optimal threshold values of 61%, 76%, and 96%, to the four, three, and two-category classifier confusion matrices.
(TIF)

**S9 Fig. A healthy control dataset was processed using our four-category classifier operating at 285 nm.** Top: Sample composition estimates are shown as a function of confidence

threshold, demonstrating that the approximately 0.5% raw FPR can be reduced by rejecting cells with low classifier confidence. Bottom: The number of cells kept (blue) and rejected (red) by the thresholding process are plotted as a function of threshold value.
(TIF)

**S10 Fig. Slice-dependent four-category classification performance arranged as a confusion matrix.** All panels spatially correspond to confusion matrix elements from Fig 2, and display relative changes in matrix values as a function of defocus. Diagonal matrix entries (true positives) corresponding to all three parasite life cycles stages tend to improve with global best focus, while many off-diagonal entries (false negatives / false positives) see reductions at the global best focus. However, exceptions include ring false positives (second row, first column) and trophozoite/schizont cross-identification.
(TIF)

**S11 Fig. Multi-wavelength image analysis pipeline steps.** 1. Load data structures (UV scope) or raw images (commercial scope). 2. Apply slice selection. 3. Register wavelength channels using affine transformation. 4. Export pre-processed images. 5. Do semantic segmentation on the master wavelength/slice using ResNet-50. 6. Apply instancing algorithm to all wavelengths/slices, exporting RBC instances to disk. 7. Classify all RBC instances for only the master wavelength and slice. 8. Apply the resulting labels to all the corresponding instances in the other slices/channels. 9. Export the lowest-confidence subset of the machine-classified images for human annotation. 10. Import the results of human annotation and propagate labels to all the other channels/slices. 11. Train new classifiers with the high confidence dataset.
(TIF)

**S12 Fig. Affine transformations were performed for image registration between different wavelengths.** Left: overlay of two raw images from the UV scope, acquired at 285 nm (green) and 565 nm (pink). Right: overlay of the same images after image registration was completed. The sample in the images consists of fixed *E. Coli* cells (rods) mixed with 1 μm beads (circles).
(TIF)

**S13 Fig. Point Spread Function (PSF) analysis of both microscopes.** PSFs were acquired in the brightfield modality by imaging 100 nm gold nanoparticles adhered to the surface of a 0.25 mm thick quartz coverslip. All images in a) and b) are shown at the same spatial scale such that sample plane distance is directly comparable between images. a) Commercial $40 \times /1.3$ microscope PSFs at 365, 405, and white LED. PSFs display substantial spherical aberration as evidenced by elongated axial depth of focus and broader in-plane width, likely resulting from a mismatch in coverslip thickness. b) UV Scope PSFs acquired at 285, 365, and 565 nm, exhibiting diffraction-limited widths, both in-plane and axially. c) Summary of lateral PSF widths, as approximated by Gaussian fits. d) Summary of axial widths, as approximated by Gaussian fits.
(TIF)

**S14 Fig. Summary of resolution, depth of focus, and sampling characteristics.**
(TIF)

**S15 Fig. Color-focus arrays for many example RBCs at various stages of infection.** For all panels, the color/focus layout is identical to Fig 4. Examples of all four categorized life cycle stages are shown.
(TIF)

**S16 Fig. Confusion matrices for classifiers trained at each wavelength on our commercial microscope.** Left: 365 nm, Middle: 405 nm, Right: Visible light LED. All classifiers were

trained on the identical set of RBC instances, using images acquired each at their respective wavelength (multiwavelength dataset, see Table 1).
(TIF)

**S17 Fig. Parameterization of the false-positive rate from a healthy control sample imaged on our commercial microscope at 405 nm.** The upper plot displays the sample composition error for all four stages as a function of confidence threshold. Data processed were from a healthy donor sample, uninfected by *Plasmodium*. In particular, the FPR for rings at 50% confidence threshold is substantially lower than the results reported in Fig 5. We explain this by noting this particular healthy control sample exhibited very few echinocytes, to which we attribute the low basal FPRs as compared to Fig 5.
(TIF)

**S18 Fig. Total loss function (training + validation) during Faster-RCNN training.**
(TIF)

**S19 Fig. Confusion matrices for four, three, and two-category classifiers at 285 nm tested on a random sub-partition of pooled titration data, which as a whole contained a low echinocyte fraction.** As compared with Fig 2, the FPR for rings is reduced presumably due to fewer cells containing anomalous morphology, and is in closer agreement to the result extracted by least squares fit to the titration data in Fig 8. The random sub-partition was configured to contain overall parasite prevalence similar to Fig 2.
(TIF)

**S20 Fig. Overall wiring diagram of the UV Scope.** The microscope hardware was controlled by a centralized PC running Windows 10. All the various hardware devices were controlled by a combination of standard and custom Matlab classes, as described in S22 Fig. Other specific hardware configurations are possible with minor changes to the software.
(TIF)

**S21 Fig. Four-channel relay multiplexer.** The LED multiplexer uses a four-channel relay (Denkovi DAE-RB/Ro4-JQC-5V, Byala, Bulgaria) to route the current generated by a Thorlabs LED current driver (LEDD1B) to one of four LEDs. The relay board was mounted inside a box to house the wired connections between relays. The box received a three-pin signal (plus GND). The first relay was used as an overall ON/OFF switch, while the remaining three were arranged in a binary tree to route current from the driver to any of the four LEDs.
(TIF)

**S22 Fig. Software architecture.** A set of custom Matlab classes were used to configure and execute multi-dimensional image acquisitions. Multiplexed LED illumination was driven by a data class which maps maximum allowable LED currents to control voltages, sending a digital address to the relay multiplexer for LED selection, and an analog voltage representing percent power. The MDEngine class implements the conversion from acquisition parameters to active control of the UVScope, including real-time focus tracking across large sample areas, in order to keep focal stacks centered on the sample. All microscope control software can be found at: https://github.com/czbiohub/UVScope-control.
(TIF)

## Acknowledgments

The authors would like to acknowledge Shalin Mehta, Loic Royer, Bin Yang, and Kevin Yamauchi for useful discussions related to construction of the microscope and its software

design, and Robert Puccinelli for building the relay-based LED multiplexer. We would also like to thank Madhura Raghavan from the DeRisi lab for assistance with manual parasite counting, and Joshua Batson for discussions related to statistics and deep learning. We are grateful to the late Prof. James Pawley for providing the quartz microscope objective and for insightful written communications on ultraviolet imaging.

## Author Contributions

**Conceptualization:** Paul Lebel, Joseph DeRisi, Rafael Gómez-Sjöberg.

**Data curation:** Paul Lebel, Rebekah Dial, Venkata N. P. Vemuri.

**Formal analysis:** Paul Lebel, Venkata N. P. Vemuri.

**Investigation:** Paul Lebel, Rebekah Dial, Valentina Garcia, Joseph DeRisi, Rafael Gómez-Sjöberg.

**Methodology:** Paul Lebel, Rebekah Dial, Venkata N. P. Vemuri, Valentina Garcia, Rafael Gómez-Sjöberg.

**Project administration:** Paul Lebel, Joseph DeRisi, Rafael Gómez-Sjöberg.

**Software:** Paul Lebel, Venkata N. P. Vemuri.

**Supervision:** Joseph DeRisi, Rafael Gómez-Sjöberg.

**Validation:** Paul Lebel, Valentina Garcia.

**Visualization:** Paul Lebel, Venkata N. P. Vemuri.

**Writing – original draft:** Paul Lebel.

**Writing – review & editing:** Paul Lebel, Rebekah Dial, Venkata N. P. Vemuri, Valentina Garcia, Joseph DeRisi, Rafael Gómez-Sjöberg.

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
