## [Decision Letter · Decision Letter 0]

21 Apr 2021

Dear Dr. Lebel,

Thank you very much for submitting your manuscript "High-performance automated classification of live P. falciparum can be achieved using ordinary brightfield microscopy: A superior alternative to labor-intensive fixation and staining." for consideration at PLOS Computational Biology.

As with all papers reviewed by the journal, your manuscript was reviewed by members of the editorial board and by several independent reviewers. In light of the reviews (below this email), we would like to invite the resubmission of a significantly-revised version that takes into account all the reviewers' comments.

In relation to comments by reviewer one in relation to the AI, I would encourage to make these points in the discussion of the novelty vs. the implementation novelty of the approach and how it relates to the current state-of-the-art in AI for medical image analysis. In relation to reviewer 2, I do expect a well sustained revision and comments of the points highlighted. I would also to have a clear discussion on the complexity of the problem in relation to the context of (in vitro specimens vs. clinical diagnosis specimens) as these pose different challenges. I would consider to revisit the title from "High-performance automated classification of live P. falciparum can be achieved using ordinary brightfield microscopy: A superior alternative to labor-intensive fixation and staining." "Stain-free high-performance automated classification of live P. falciparum using brightfield microscopy" AND also consider be specific where you want to make the claim, eg clinical vs. biomedical use. 

We cannot make any decision about publication until we have seen the revised manuscript and your response to the reviewers' comments. Your revised manuscript is also likely to be sent to reviewers for further evaluation.

Sincerely,

Delmiro Fernandez-Reyes, M.D., M.Sc., D.Phil

Guest Editor

PLOS Computational Biology

Alice McHardy

Deputy Editor

PLOS Computational Biology

Reviewer's Responses to Questions

**Comments to the Authors:**

Reviewer #1: The authors describe the development of an automated method and algorithms for identifying and counting red blood cells infected with P. falciparum. The stunning thing is that it can be done on life parasites. I cannot really comment much on the technical aspects of the method development and how much knowledge gain this actually represents, but the description is exceptionally detailed. I have no reason to believe that the method is not performing as described by the authors.

What I can comment on however is the claim that this method would make standard malaria microscopy obsolete. I think this is quite grossly overstated. Right now, as far as I understand, the automated method is able to achieve limit of detection of around 0.1% parasitemia (abstract). This is nowhere near the limit of detection achieved by standard light microscopy for malaria diagnosis, which uses thick blood films and has a limit of detection of around 10 (5-20) parasites per microliter of blood [1, 2]. Even though it is not entirely appropriate, for illustrative purposes, we can convert this to parasitemia based on the assumption that one microliter of blood has around 5 million red cells. Thus, the LOD of gold standard, expert light microscopy is on the order of around 0.0002% parasitemia and thus literally a thousand-fold better than the automated method described. So the new method would have to be able enumerate several million cells in around 30 minutes to be able to compete with standard light microscopy in terms of time to diagnosis and limit of detection. Given the complexity of the instrumentation and computational resources (in particular in view of resource limited settings where malaria is important), it is hard to imagine this to rapidly transition to something used for malaria diagnosis in Africa.

Nevertheless, given the rapid gains made with AI and computational power, it may be possible that in a few years time the required numbers of cells can be enumerated. As such, the question whether to accept (with major revisions) or reject this manuscript is probably about whether the AI development and advance made in technology represent a knowledge gain that is significant enough to warrant publication, as this has no impact on clinical or laboratory practice in the near future. I am not able to judge on this part but can highlight that there are other automated microscopy methods that have been proposed in the past so the idea is certainly not new and there are many papers out there (I just give one example here).[3] But doing it with life cells from fresh blood is certainly a step forward and I feel that the presentation of the method that the authors put together in their manuscript is excellent.

1. World Health Organisation: Basic malaria microscopy. 2nd ed edition. Geneva. (2010).

2. Schneider, P. et al. (2004) Quantification of Plasmodium falciparum gametocytes in differential stages of development by quantitative nucleic acid sequence-based amplification. Mol Biochem Parasitol 137 (1), 35-41.

3. Ross, N.E. et al. (2006) Automated image processing method for the diagnosis and classification of malaria on thin blood smears. Med Biol Eng Comput 44 (5), 427-36.

Reviewer #2: the review is given as an attached file

**Have all data underlying the figures and results presented in the manuscript been provided?**

Reviewer #1: Yes

Reviewer #2: Yes

PLOS authors have the option to publish the peer review history of their article (what does this mean?). If published, this will include your full peer review and any attached files.

Reviewer #1: No

Reviewer #2: No
---

## [Decision Letter · Decision Letter 1]

7 Jul 2021

Dear Dr. Lebel,

We are pleased to inform you that your manuscript 'Label-free imaging and classification of live P. falciparum enables high performance parasitemia quantification without fixation or staining.' has been provisionally accepted for publication in PLOS Computational Biology.

Best regards,

Delmiro Fernandez-Reyes, M.D., M.Sc., D.Phil

Guest Editor

PLOS Computational Biology

Alice McHardy

Deputy Editor

PLOS Computational Biology

Reviewer's Responses to Questions

**Comments to the Authors:**

Reviewer #2: The authors have addressed all my comments with honesty and now provide the missing details through the supplementary figures S13 et S14. These really help to better understand their discussion about the spatial resolution and makes the optical part of the paper a lot clearer.

Given the results provided and the effort made for taking into account the different comments, I think the paper deserves to be published.

**Have the authors made all data and (if applicable) computational code underlying the findings in their manuscript fully available?**

Reviewer #2: Yes

PLOS authors have the option to publish the peer review history of their article (what does this mean?). If published, this will include your full peer review and any attached files.

Reviewer #2: No

---

## [Editor Report · Acceptance letter]

4 Aug 2021

PCOMPBIOL-D-20-02077R1 

Label-free imaging and classification of live *P. falciparum* enables high performance parasitemia quantification without fixation or staining.

Dear Dr Lebel,

I am pleased to inform you that your manuscript has been formally accepted for publication in PLOS Computational Biology. Your manuscript is now with our production department and you will be notified of the publication date in due course.

With kind regards,

Andrea Szabo
